# Assessing representativity of NH$_3$ measurements influenced by boundary-layer dynamics and turbulent dispersion of a nearby emission source

Ruben B. Schulte[1], Margreet C. van Zanten[1,2], Bart J.H. van Stratum[1], and Jordi Vilà-Guerau de Arellano[1]

[1]Wageninen University & Research, P.O. Box 47, 6700, AA Wageningen, the Netherlands
[2]National Institute for Public Health and the Environment (RIVM), Antonie van Leeuwenhoeklaan 9, 3721, MA Bilthoven, the Netherlands

**Correspondence:** Ruben Schulte (ruben.schulte@wur.nl)

**Abstract.**

This study presents a fine-scale simulation approach to assess the representativity of ammonia (NH$_3$) measurements in proximity of an emission source. Close proximity to emission sources (< 5 km) can introduce a bias in regionally representative measurements of the NH$_3$ molar fraction and flux. Measurement sites should therefore be located a significant distance from emission sources, but such requirements are poorly defined and can be difficult to meet in densely agricultural regions. This study presents a consistent criterion to assess the regional representativity of NH$_3$ measurements in proximity of an emission source, calculating variables that quantify the NH$_3$ plume dispersion using a series of numerical experiments at a fine resolution (20 m). Our fine-scale simulation framework with explicitly resolved turbulence enables us to distinguish between the background NH$_3$ and the emission plume, including realistic representations of NH$_3$ deposition and chemical gas-aerosol transformations. We introduce the concept of blending-distance, based on the calculation of turbulent fluctuations, to systematically analyze the impact of the emission plume on simulated measurements, relative to this background NH$_3$. We perform a suite of systematic numerical experiments for flat homogeneous grassland, centered around the CESAR Observatory at Cabauw, to analyze the sensitivity of the blending-distance, varying meteorological factors, emission/deposition and NH$_3$ dependences. Considering these sensitivities, we find that NH$_3$ measurements at this measurement site should be located at a minimum distance of 0.5 - 3.0 km and 0.75 - 4.5 km from an emission source, for NH$_3$ molar fraction and flux measurements, respectively. The simulation framework presented here can easily be adapted to local conditions and paves the way for future ammonia research to integrate simulations at high spatio-temporal resolution with observations of NH$_3$ concentrations and fluxes.

## 1 Introduction

Excess atmospheric nitrogen leads to an increased public health risk, through the formation of particulate matter, and causes environmental damage, as nitrogen deposition leads to eutrophication, ecosystem acidification and shifts in climate change (Erisman and Schaap, 2004; Sutton et al., 2008; Behera et al., 2013; Erisman et al., 2013; Smit and Heederik, 2017). There can

be serious societal consequences when nitrogen deposition critical loads are exceeded, as is the case in the Netherlands where the nitrogen crisis threatens the Dutch environment and economy (Stokstad, 2019). Atmospheric ammonia ($NH_3$) plays a key role in this process, mainly originating from agricultural activities and accounting for two-thirds of all nitrogen deposition in the Netherlands between 2005 and 2016 (Wichink Kruit and van Pul, 2018).

It is therefore important to have a network of $NH_3$ concentration and deposition measurements, used for model validation and (trend) monitoring (Wichink Kruit et al., 2021). For these purposes, the measurement sites in such a network must be representative for a larger region. One requirement for such regional measurement sites is to be located at sufficient distance from local $NH_3$ sources, as local emissions introduce a bias in the observations (EMEP/CCC, 2001; Wichink Kruit et al., 2021). Positioning measurements sites at sufficient distance from local sources is a challenge in densely agricultural areas like the Netherlands and regions all across the world with intensive livestock farming, e.g. North-West Germany, the province of Lerida in Spain, the state of North-Carolina in the USA or the Hai River Basin in China.

The emitted $NH_3$ is transported and mixed within the convective boundary layer (CBL) through turbulent dispersion. The field of turbulent plume dispersion is extensively researched using both observations and turbulent resolved models. However, such studies typically focus on concentration peaks of highly toxic/flamable gasses (Mylne and Mason, 1991; Ardeshiri et al., 2021; Cassiani et al., 2020), quantification of the emission strength and position (Shah et al., 2020; Ražnjević et al., 2022) or on statistical descriptions of the emission plume (Barad, 1958; Dosio et al., 2003; Vrieling and Nieuwstadt, 2003; Dosio and Vilà-Guerau de Arellano, 2006), typically used in chemistry transport models, e.g. OPS (Sauter et al., 2018), LOTOS-EUROS (Schaap et al., 2008) or EMEP MSC-W (Simpson et al., 2012). These transport models typically operate with resolutions at kilometer scale (1 - 50 km) and parameterized turbulence, making them unsuitable to study the impact of local $NH_3$ sources on nearby measurement sites at the subkilometer scale.

Furthermore, plume dispersion studies generally focus on chemically inert gasses, e.g. methane (Shah et al., 2020; Ražnjević et al., 2022). Ammonia is highly reactive: surface-atmosphere exchange and chemical gas-aerosols transformations play an important role in the $NH_3$ budget (Fowler et al., 1998; Van Oss et al., 1998; Nemitz et al., 2004; aan de Brugh et al., 2013; Behera et al., 2013; Shen et al., 2016; Schulte et al., 2021). Additionally, ammonia emissions in densely agricultural areas are released and mixed into a background concentration, a result of long range transport of $NH_3$ (10-100 km). Yearly averaged background concentrations can vary from 1-2 $\mu$g m$^{-3}$ (e.g in coastal regions) up to up to tens of $\mu$g m$^{-3}$ in regions with intensive agricultural activity, which is the focus on this study (van Zanten et al., 2017).

In this study, we investigate the impact of a typical ammonia emission source on the regional representativeness of $NH_3$ concentration and flux measurements. The novelty of our approach is twofold:

- The use of a fine-scale Large-Eddy Simulation (LES) model with explicitly resolved turbulence at a very high spatio-temporal resolution (10-100 m and 10 s - 1 min).

- Inclusion of realistic representations of surface-atmosphere exchange, chemical gas-aerosol transformations and a background ammonia concentration.

Following this approach, we combine fine-scale simulations, where turbulence is explicitly resolved, with concepts of theory on turbulent emission plume dispersion and translate this knowledge to practical applications for the measurement community. The aim is to carry out a systematic analysis on how meteorological factors, including boundary-layer dynamics, deposition, chemical transformation and model resolution influence the relationships between emission and receptor. To this end, we introduce and analyze the concept of a blending-distance (BD), i.e. the horizontal distance at which the emission plume can

be considered well-mixed with respect to the background $NH_3$. With the concept of blending-distance, we aim to provide an estimate of the minimum required distance from a typical $NH_3$ emission source for regionally representative measurements.

## 2 Methodology

### 2.1 $NH_3$ turbulent dispersion in DALES

To understand the variations of the $NH_3$ budget due to turbulence and heterogeneous sources and sinks of ammonia, our

approach is two folded: (a) explicit simulation of processes that govern turbulent dispersion and mixing of $NH_3$ and (b) identifying their individual contributions to the $NH_3$ molar fraction and surface-atmosphere exchange. For the former, we use the large-eddy simulation technique with a high resolution to solve explicitly turbulence. To this end, we conduct our numerical experiments using a modified version of the Dutch Atmospheric Large-Eddy Simulation (DALES) version 4.2 (Heus et al., 2010; Ouwersloot et al., 2017), with the original v4.2 freely available online (http://doi.org/10.5281/zenodo.3759193). DALES

explicitly resolves processes at scales ranging from hundred meters to kilometers, using filtered Navier-Stokes equations with the Boussinesq approximation. The filter size is generally equal to the grid size of the simulations, with subfilter-scale processes being parameterized using one-and-a-half-order closure. The numerical experiments presented here are performed using a 20 m x 20 m x 5 m grid for a 10 km x 4.8 km x 3 km domain (500 x 240 x 600 grid points). Atmospheric $NH_3$ is added to DALES as a passive scalar in ppb, of which the spatial evolution is solved simultaneously with the thermodynamic variables.

The boundary conditions for scalars and meteorological variables are periodic, unless stated otherwise.

The atmospheric ammonia budget is further governed by surface-atmosphere exchange and chemical gas-particle transformations (Schulte et al., 2021). We use a simplified, yet realistic, approach in our representation of these processes. $NH_3$ surface-atmosphere exchange is modeled by a constant homogeneous deposition of 0.045 ppb m s$^{-1}$ (about 0.032 $\mu$g m$^{-2}$s$^{-1}$), representative for the observed yearly average $NH_3$ dry deposition in the Netherlands (https://www.rivm.nl/stikstof/meten/

drogedepositieNH3; Stolk et al., 2014).

The representation of the chemical gas-aerosol transformations follows the approach of the OPS model: applying a percentage per hour change in the molar fraction of gaseous $NH_3$ to the whole domain (van Jaarsveld, 2004). This simplified yet realistic representation of chemistry as a net removal process will reduce the reach of the emission plume. However, the model is unable to resolve potential non-linear effects of turbulent mixing on the chemical reaction rate within the plume. Turbulent

dispersion of the emission plume is characterized by macromixing (meandering) and micromixing (in-plume mixing) (Vilà-Guerau de Arellano et al., 1990; Galmarini et al., 1995). The former is mainly carried out by large-scale turbulent eddies and is related to the average dispersion of the plume. Micromixing is carried out by turbulent eddies smaller than the plume and is

related to the fluctuations of $NH_3$ and its chemical reactants. The reaction rate can slow down close to the emission source, as macromixing is the dominant dispersion process here and little micromixing occurs to supply chemical reactants from outside the plume. The extend at which turbulent mixing can limit the chemical reactions within the plume depends on the ratio of the turbulent time scales and the time scale of chemistry (Damköhler number) (Galmarini et al., 1995; Meeder and Nieuwstadt, 2000). When the time scales of chemistry are similar to the turbulent time scales, as is the case for ammonia (aan de Brugh et al., 2013), the reduction in the chemical reaction rate close the the source can be significant (Vilà-Guerau de Arellano et al., 2004).

Special attention is placed on the representation of the one $NH_3$ emission source in our domain, representing a dairy barn. Agricultural activity accounts for over 90% of the $NH_3$ emissions in the Netherlands and the European Union (Anys et al., 2020; Vonk et al., 2020; van Bruggen et al., 2021). Dairy farms account for approximately 50% of these agricultural $NH_3$ emissions, with approximately 15.000 farms with about 100 cows each on average in the Netherlands (van der Peet et al., 2018; WUR, 2021). A typical cubicle stable for 80 cows has a yearly emission of about 800 kg $NH_3$ year[-1] and requires 10 m$^2$ per cow (800 m$^2$ in total) (Remmelink et al., 2020, Table 10.19; RIVM, 2021, type A1). Contrary to the closed off and air filtered housing for pigs and chickens, a dairy barn is open and the ammonia-rich air can freely escape. Therefore, we are able to represent a typical 80 dairy cow barn as a surface emission source (Theobald et al., 2012) with an emission flux of 45 ppb m s[-1] (about 32 $\mu$g m$^{-2}$s$^{-1}$) over an area of 800 m$^2$.

We identify the individual contributions of ammonia sources to the $NH_3$ molar fraction and surface-atmosphere exchange, with each source of $NH_3$ represented by a unique scalar. In this study, these sources are identified as a background molar fraction ($NH_{3,bg}$) and the $NH_3$ emission plume ($NH_{3,plume}$) from a surface emission source. The sum of these two unique scalars represents the total atmospheric ammonia ($NH_{3,total}$), as would be observed by in-field observations. Here, we modify DALES v4.2 to force the $NH_{3,plume}$ molar fraction to zero at both x-edges of the domain (west and east), preventing circulation of the emission plume in x-direction.

Further modifications to DALES v4.2 are made to include the remaining processes governing the variability of the atmospheric ammonia budget. The scalar surface flux ($F_{total}$), representing surface atmosphere exchange, is divided between a flux acting on the background scalar ($F_{bg}$) and another flux acting on the emission plume scalar ($F_{plume}$). The magnitude of these two fluxes is weighted by their respective molar fractions ($NH_{3,bg}$ and $NH_{3,plume}$) relative to the total $NH_3$ molar fraction, e.g. $F_{bg} = \frac{NH_{3,bg}}{NH_{3,total}} F_{total}$ for $NH_{3,bg}$.

The final modification adds an additional term to the change in the scalar molar fraction ($\frac{S}{dt}$). This modified change in the scalar molar fraction reads: $\frac{dS}{dt} + \frac{R_{chem}}{3600} S$, with $R_{chem}$ representing the gain/loss rate in % hour[-1] and subscript $S$ representing the scalar molar fraction, which can be substituted by either $NH_{3,plume}$ or $NH_{3,bg}$. The modified DALES v4.2 code used in this study is also freely available online (http://doi.org/10.4121/19869478).

## 2.2 Numerical experiments

We simulate the meteorological conditions observed on 8 May 2008 at the Ruisdael CESAR Observatory (https://ruisdael-observatory. nl/cesar/) at Cabauw in the Netherlands (51.971ºN, 4.927ºE), as described by aan de Brugh et al. (2013) and Barbaro et al.

(2014, 2015). The supersite, with a 213 m high mast, is located on flat (agricultural) grassland with an average height of 0.1 m and the surface elevation changes are at most a few meters over 20 km. 8 May 2008 is selected as it is widely studied and includes measurements of the $NH_3$ molar fraction. In May 2008, the intensive observational campaign IMPACT/EUCAARI was
125 held, which included ammonia concentration measurements by a MARGA system (aan de Brugh et al., 2012; Mensah et al., 2012) and several additional meteorological variables, including vertical profiles and radiosondes (Kulmala et al., 2011). The meteorology of this day is described in detail by Barbaro et al. (2014), where the experiment is called CESAR2008. Figures 2 and 3 by Barbaro et al. (2014) show vertical profiles and time series of, among other variables, potential temperature, specific humidity, surface fluxes and boundary layer height. The case can be characterized as typical clear-sky, fair-weather conditions
with an absence of large-scale heat advection. The model is initialized following the conditions as described by Barbaro et al. (2014) and the initial and prescribed meteorological values of the experiment can be found in Barbaro et al. (2014) Table 1.

In the morning, a 1500 m residual layer leads to an overshooting of the boundry layer height around 10:30 CEST, up to roughly 1800 m. In the afternoon (12:30 – 17:00 CEST), CBL growth is weak and the thermodynamic conditions remain relatively constant (Barbaro et al., 2014). Therefore, we only study the turbulent dispersion in the afternoon, when the impact
of boundary layer dynamics on the NH3 budget is minimal. The wind speed is moderate at 5.5 to 7 m s$^{-1}$ in the afternoon, resulting in strong shear production near the surface and a strong momentum entrainment at the CBL top. The convective time scale ($\tau$) in the afternoon is typical for convective fair-weather conditions, increasing from 18 to 27 minutes between 12:30 and 17:00 CEST. The Monin-Obukhov length fluctuates around approximately -50 m.

The numerical experiments are split into three phases: the meteorological spin-up phase, the buffer phase and the analysis
phase. During the meteorological spin-up, 8:00 – 12:30 CEST, the ammonia surface-atmosphere exchange and chemical transformations are not active. These processes are activated at the start of the buffer phase, from 12:30 – 14:00 CEST. Entrainment is still an important factor until around 13:00 CEST, causing large fluctuations of the $NH_3$ molar fraction (> 4 ppb) as will be discussed in Sect. 3.1. The CBL is considered well-mixed around 13:00 CEST, but we extend the buffer phase with one more hour. We do so to minimize impact of earlier entrainment on the one-hour moving average used to calculate statistics
during the analysis phase. The analysis phase therefore starts at 14:00 CEST until the collapse of the CBL around 17:00 CEST. The analysis phase is the focus of this study and when we analyze the impact of the emission plume on (simulated) point measurements of the $NH_3$ concentration and flux.

### 2.3 Quantifying the emission plume impact on $NH_3$ measurements

Inspired by the plume observation study by Mylne and Mason (1991), we introduce three variables to assess the presence of
150 the emitted $NH_3$ plume and relevance of the plume fluctuations to nearby observations. These variables, intermittency factor (I), fluctuation intensity (fI) and $NH_3$ flux (F), are all defined by fluctuations in the $NH_3$ molar fraction. Fluctuations in the $NH_3$ molar fraction result from turbulent mixing of differences in $NH_3$, caused by local sinks and sources. $NH_3$ fluctuations are therefore found in the background molar fraction as a result of ammonia-poor air near the surface (deposition) and top of the CBL (entrainment). $NH_3$ fluctuations are further enhanced in proximity of surface heterogeneous surfaces. A strong local
emission source (e.g. a dairy barn) as presented in this study, will cause an emission plume as the enhanced $NH_3$ molar fraction

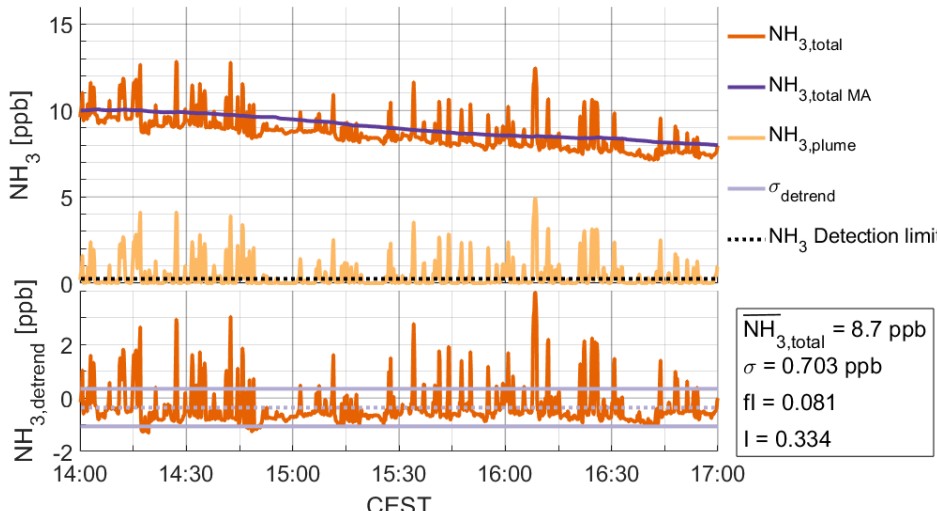

**Figure 1.** Top panel shows 10 s time series of NH₃,total (orange) and NH₃,plume (yellow) during the analysis phase, at 250 m from the emission source. The detrended NH₃,total (orange) is shown in the bottom panel. Fluctuation intensity and intermittency are calculated following Eq. 4 and 1 respectively, based on the mean NH₃,total, standard deviation (light purple) and NH₃ detection limit (dotted black).

is mixed with the background molar fraction through turbulent mixing. Turbulent models like DALES explicitly resolve this turbulent mixing at high spatial-temporal resolution and can provide valuable information in the interpretation of in-field observations where surface heterogeneity plays an important role.

We first introduce the intermittency factor (I) to quantify the detectability of the emission plume. Intermittency is defined as the proportion of time during which the plume molar fraction is above the detection limit of instruments typically used to measure atmospheric ammonia, as seen in Fig. 1 and Eq. 1, where N is the number of time steps.

$$I = \frac{1}{N} \sum_{i=1}^{N} \begin{cases} 1, & \text{if } NH_{3,plume}(i) \geq 0.25 ppb \\ 0, & \text{if } NH_{3,plume}(i) < 0.25 ppb \end{cases} \tag{1}$$

Note that the intermittency is calculated for each individual grid point during the analysis window (14:00 - 17:00 CEST) at 10 s temporal resolution. We set the NH₃ detection limit at 0.25 ppb, similar to the detection limit of the miniDOAS instrument used in the Dutch ammonia monitoring network (Berkhout et al., 2017). The concept of intermittency cannot be applied to NH₃,bg or NH₃,total, as the background molar fraction always exceeds 0.25 ppb in our numerical experiments, which would result in an intermittency of 1. We therefore only calculate the intermittency for NH₃,plume to analyze the detectability of the emission plume.

The second variable, fluctuation intensity (fI), determines the magnitude of the NH₃ fluctuations, i.e. NH₃ standard deviation ($\sigma_{NH_3}$), relative to the mean NH₃ molar fraction ($\overline{NH_3}$). Fluctuation intensity is defined following Eq. 2:

$$fI = \frac{\sigma_{NH_3}}{\overline{NH_3}} \tag{2}$$

The fluctuation intensity quantifies the level of turbulent mixing. High fI indicates that there are large fluctuations in the measured $NH_3$ which can introduce a positive bias in measurements. In the field of plume dispersion, high fI is found close to the source where plume meandering dominates the mixing process (Dosio and Vilà-Guerau de Arellano, 2006), or at the
175 edge of the emission plume as a result of lateral entrainment of air from outside the plume (Mylne and Mason, 1991; Gailis et al., 2007; Ražnjević et al., 2022). When analyzing the fluctuation intensity of $NH_{3,total}$, we have a consistent reference for the fluctuation intensity in $NH_{3,bg}$. Comparing the fI for the total ammonia ($fI_{total}$) to the fI for the background ammonia ($fI_{bg}$), enables us to quantify the relative impact of the emitted $NH_3$ plume to simulated measurement. When $fI_{total}$ is of the same order of magnitude as $fI_{bg}$, we consider the emission plume indistinguishable from the background $NH_3$, i.e. the plume is well
mixed. Note that for $NH_{3,\,plume}$, the average $NH_3$ concentration is (very close to) zero outside the emission plume, which could lead to infinitely large fluctuation intensity following Eq. 2. Therefore, fI is only calculated inside the plume, using an arbitrary requirement of $\overline{NH}_{3,plume} > 10^{-5}$ ppb.

Fig. 1 shows a downward trend in $NH_{3,bg}$ and $NH_{3,total}$, resulting from surface deposition and the loss by chemical gas-aerosol transformations. To minimize the impact of this downward trend on $\sigma_{NH_3}$, we detrend the simulated molar fraction by subtract-
185 ing a 1 hour leading moving average ($NH_{3,MA}$), following Eq. 3 and shown in Fig. 1. The detrended molar fraction ($NH_{3,detrend}$) is assumed to only represent turbulent fluctuations and is used to calculate the standard deviation to derive fluctuation intensity. By using $NH_{3,detrend}$ to calculate $\sigma_{NH_3}$, the fluctuation intensity follows from Eq. 4.

$$NH_{3,detrend} = NH_3 - NH_{3,MA} \tag{3}$$

$$
\begin{aligned}
fI &= \frac{\sigma_{NH_3}}{\overline{NH_3}} \\
&= \frac{\sqrt{\frac{1}{N-1}\sum_{i=1}^{N}|NH_{3,detrend} - (\overline{NH_{3,detrend}})|^2}}{\overline{NH_3}}
\end{aligned} \tag{4}
$$

Finally, we introduce the 30 minute $NH_3$ flux, studied to mimic the in-field ammonia eddy covariance flux measurements and calculated following Eq. 5. The flux presented in this study is the average 30 minute flux, for each individual grid point, over the analysis phase between 14:00 and 17:00 CEST.

$$F_{NH_3} = \overline{NH_3' w'} \tag{5}$$

## 2.4 The concept of blending-distance

We use the fluctuation intensity and flux to quantify the impact of the emission plume on the simulated $NH_3$ molar fraction and flux measurements, by introducing the concept of blending-distance. The blending-distance is based on the percentage change

(PC$_X$) in the simulated NH$_3$ measurements resulting from the emission plume, i.e. the percentage change between NH$_{3,\text{total}}$ and NH$_{3,\text{bg}}$. PC$_X$ is calculated following Eq. 6, where X can be substituted by either fI or F.

$$PC_X = |\frac{X_{total} - X_{bg}}{X_{bg}}| * 100\% \tag{6}$$

Based on this percentage change, we define a threshold for which we assume that the impact of the emission plume is negligible. The blending-distance (BD$_X$, is defined as the maximum distance at which PC$_X$ drops below the threshold level (e.g. PC$_X$ < 25%), following Eq. 7.

$$BD_X = max(\ dist(\ PC_X < threshold\ )\ ) \tag{7}$$

In this study, we present blending-distances based on an arbitrary set of threshold levels, ranging from 5% to 50%.

The concept of blending-distance is applied to the fluctuation intensity (BD$_{\text{fI}}$) and the NH$_3$ flux (BD$_F$) to quantify the impact on the simulated NH$_3$ measurements of NH$_3$ molar fraction and flux respectively. For context, we also present the intermittency in Sect. 3.2 to quantify the detectability of the plume.

### 2.5  Blending-distance sensitivity

A key aspect of the study is to determine the sensitivity of the concept of the blending-distance to variations in meteorological and NH$_3$ pollution factors, in order study the impact of each processes on the blending-distance and to identify the driving variables. We study the sensitivity of blending-distance for fluctuation intensity and NH$_3$ flux by varying the geostrophic wind speed (u$_g$), initial background molar fraction (C$_{\text{bg}}$) at the start of the analysis phase, emission strength (E), deposition strength (D), chemical conversion rate (R), simulation height (H) and model grid resolution ($\Delta$). Table 1 presents the suite of numerical experiments presented in this study. A single numerical experiment was performed for the sensitivity studies of the NH$_3$ background, emission, deposition and chemistry, each with separate scalars for NH$_{3,\text{bg}}$ and NH$_{3,\text{plume}}$. This single experiment, which does not include the variations in the geostrophic wind speed nor the high-resolution experiment, generates just under 1 TB of model output with a computational cost of about 64.000 SBU (System Billing Unit, i.e. the usage of one processor of the Cartesius supercomputer system for one hour).

The sensitivity study is structured from large-scale processes to small scale processes and modeling numerics. Starting with mesoscale processes, we vary the geostrophic wind speed to study the impact of the atmospheric stability on blending-distance, i.e. a shear or convection dominated CBL. Atmospheric stability plays a key role in turbulent mixing of local sources (emission) and sinks (entrainment and deposition), affecting both the fluctuations in the background molar fraction and the mixing of the emission plume (Dosio et al., 2003). Next, we study the sensitivity of BD to different levels of the background NH$_3$ at the start of the analysis window, representing different levels of regional NH$_3$ pollution. Additionally, varying the background levels of ammonia changes the NH$_3$ inversion at the top of the CBL, affecting the impact of entrainment. Next, the emission strength is varied, in order to study the local effect of different emission strengths.

Furthermore, we study the sensitivity of both BD$_{\text{fI}}$ and BD$_F$ to NH$_3$ deposition and the chemical gas-aerosol transformation. These are dynamic processes, i.e. experiencing clear diurnal and seasonal variability, mainly related to temperature, humidity

**Table 1.** Parameter names, symbols, reference values and their respective variations for the sensitivity study of the blending-distance, with the reference settings highlighted in bold.

| Parameters | Symobol | Reference experiment | Variations | | | | |
|---|---|---|---|---|---|---|---|
| Geostrophic wind speed | $u_g$ | 8 m s$^{-1}$ | | 2   4   6   **8**   10 | | | |
| Initial NH$_{3\,bg}$ | $C_{bg}$ | 10 | | 5   **10**   15   25 | | | |
| NH$_3$ emission strength | E | 45 ppb m s$^{-1}$ | | **45**   100   150   200 | | | |
| NH$_3$ deposition strength | D | -0.045 ppb m s$^{-1}$ | 0   -0.025   **-0.045**   -0.075   -0.0100 | | | | |
| NH$_3$ chemical conversion rate | R | 5 % hour$^{-1}$ | | 0   **5**   15   25 | | | |
| Simulated measurement height | H | 37.5 m | 7.5   12.5   ...   112.5   117.5 | | | | |
| Model resolution | Δ | 20 m x 20 m x 5 m | 10 x 10 x 2.5   **20 x 20 x 5**   50 x 50 x 15 | | | | |

and pollution levels (Wichink Kruit et al., 2010; van Zanten et al., 2010; aan de Brugh et al., 2013). Our simulation approach, with a simplified representation of deposition and chemistry, allows us to distinctly study the role of these two processes.

Finally, we study the sensitivity of BD to choices made in the numerical setup of the experiments. We vary the height of the simulated measurements. The numerical experiments are generally taken at a simulated height of 37.5 m. This is a trade-off between simulating measurements close to the surface to mimic in-field observations and the resolved turbulent kinetic energy (TKE$_{res}$) of the model. The TKE$_{res}$ at the lowest level of DALES (at 2.5 m) is zero due to the no-slip boundary at the surface (Heus et al., 2010). When we aim for a TKE$_{res}$ of 75% at all three (vertical) resolutions, we find TKE$_{res}$ of 76%, 95% and 96% for the low, middle and high resolution at 37.5 m (36.25 m for high resolution). Additionally, it is also expected that varying the measurement height will gain practical insight for in-field observations. Finally, the sensitivity of the blending-distance to changes in resolution is studied with two new numerical experiments with higher and lower resolutions of 10 m x 10 m x 2.5 m (1000 x 480 x 1200 grid points) and 50 m x 50 m x 15 m (200 x 96 x 200 grid points) respectively.

## 3   Results

### 3.1   Qualitative analysis of the NH$_3$ emission plume impact

The concept of blending-distance is based on fluctuations in the NH$_3$ molar fraction. To better understand the sources of these fluctuations, we first study the time series of a "virtual" point measurement at 250 m horizontal distance from the emission source, shown in Fig. 2a. Our simulation framework allows us to distinguish the individual contributions of NH$_{3,bg}$ (purple) and NH$_{3,plume}$ (light purple) to NH$_{3,total}$ (orange). Here we find that the large NH$_{3,total}$ fluctuations are mainly ascribed to NH$_{3,plume}$.

As discussed in Sect. 2.3, fluctuations are also found in the background molar fraction (NH$_{3,bg}$), leading to a non-zero fI$_{bg}$. Fluctuations NH$_{3,bg}$ are a result of heterogeneous turbulent mixing. In this study, the fluctuations are caused by vertical

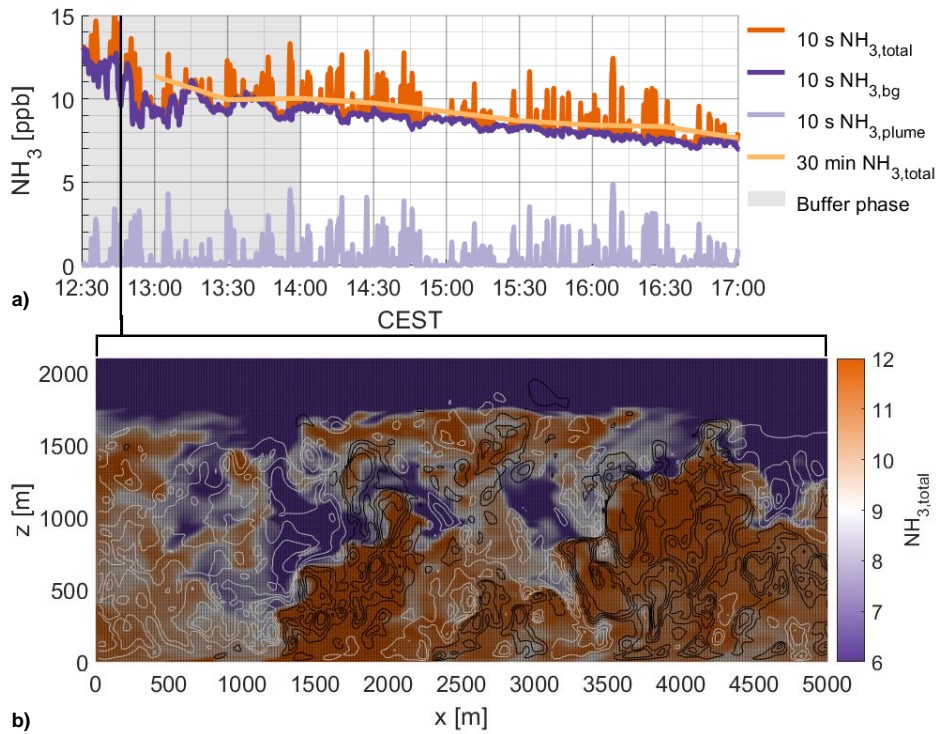

**Figure 2.** 10 s time series (a) of $NH_{3,total}$ (orange), $NH_{3,bg}$ (purple) and $NH_{3,plume}$ (light purple) during the buffer phase (grey area) and the analysis phase, taken at 250 m distance from the emission source. The large high-frequency fluctuations shown (> 4 ppb) are not captured by the 30 minute average of $NH_{3,total}$ (light orange). The vertical xz cross-section at 12:46 CEST (b), displays high spatial variability during the buffer phase in $NH_{3,total}$ (> 4 ppb) over short distances (hundreds of meters). The black/white contour lines represent upward/downward wind speed in steps of 0.5 m s$^{-1}$.

gradients only, as we use a homogeneous surface in the simulation of $NH_{3,\,bg}$. These vertical gradients are found near the surface and at the top of the CBL. At the surface, the surface-atmosphere exchange (deposition) decreases the $NH_3$ molar fraction, which results in a vertical gradient in $NH_{3,bg}$. At the top of the CBL, the vertical gradient is a result of the turbulent exchange with the free troposphere (entrainment). Fig. 2b shows that the intrusion of $NH_3$-low air masses from the free-troposphere are transported by the downdraft subsidence motions, resulting in large fluctuations in $NH_{3,bg}$ in the boundary layer. As shown in Fig. 2a, the amplitude of these fluctuations can reach 4 ppb and can last for over 5 minutes. When averaging over 30 minutes, even the large fluctuations between 12:30 and 13:15 are filtered out, but these high-frequency turbulent fluctuations could still be present in raw measurement data of high-resolution in-field observations.

Now that we understand the source of the $NH_3$ fluctuations, we take a closer look at the emission plume without any background $NH_3$. The xy plot in Fig. 3a shows low fI in the plume center ($\approx 2$) and a strong increase near the plume edges, up to fI $\approx 30$. This is echoed by the plume transects, as they shows the typical "U-shape" found for Gaussian plumes (Mylne and Mason, 1991; Gailis et al., 2007; Ražnjević et al., 2022). These high fI values at the edges of the plume are a result of

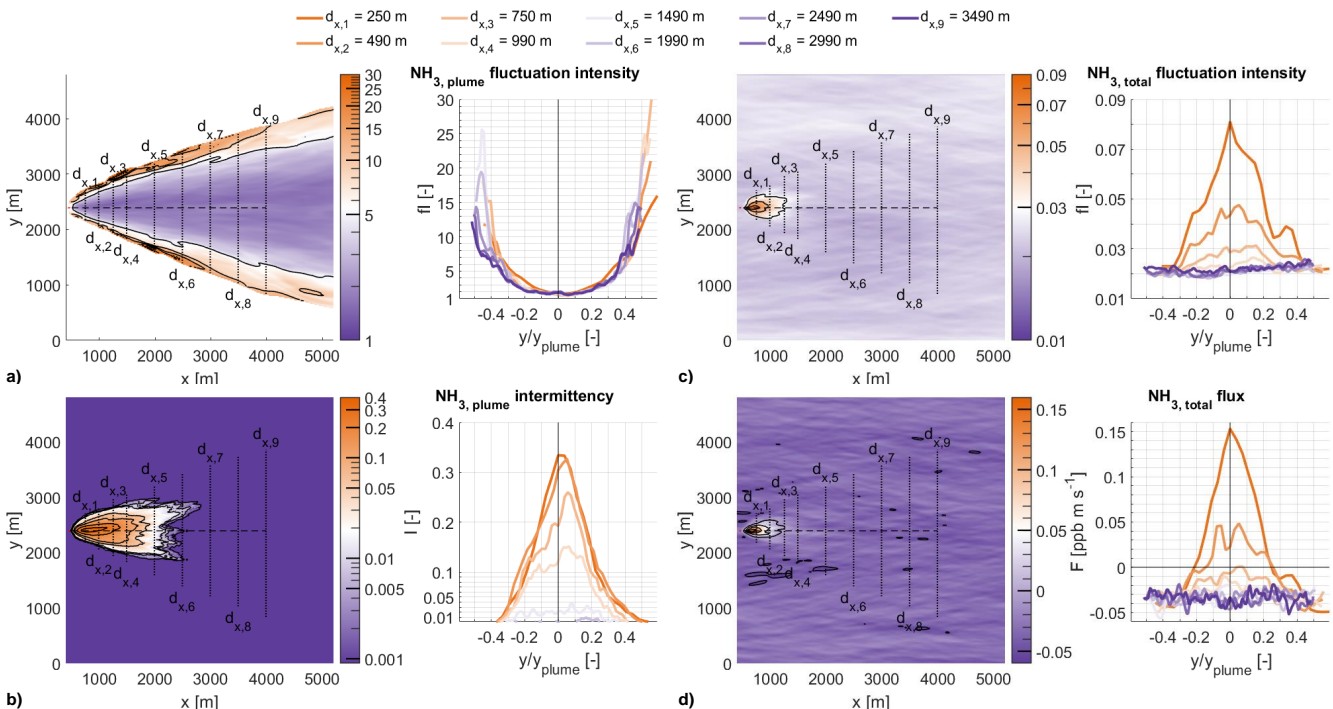

**Figure 3.** The xy cross-sections at 37.5 m with y-transects through the $NH_3$ emission plume for the $NH_{3,plume}$ fluctuation intensity (a), intermittency (b), $NH_{3,total}$ fluctuation intensity (c) and $NH_{3,total}$ flux (d). The plume transects are labeled $d_{x,1}$ to $d_{x,9}$ for increasing x-distance from the $NH_3$ emission source and normalized by plume width for $NH_{3,plume} > 10^{-5}$ ppb. The data presented is calculated during the analysis phase (14:00 and 17:00 CEST) at 37.5 m

.

very low average molar fractions combined with low intermittency. This leads to a high standard deviation, relative to the very low averaged molar fraction, at the plume edges. Without background $NH_3$, it is at the edges of the plume that in-plume lateral entrainment of ammonia-free air happens, diluting the emission plume by turbulent mixing.

    The intermittency cross-section in Fig. 3b shows that maximum I is only a little over 0.3, resulting from the meandering of the plume. Figure 3b also shows that, with an $NH_3$ detection limit of 0.25 ppb, the plume can be detected up to a distance of

about 2.0 km from the source.

    The cross-section of fI changes dramatically when analyzing $NH_{3,total}$, the sum of $NH_{3,bg}$ and $NH_{3,plume}$. With the addition of a non-zero background molar fraction, fI can be calculated over the whole domain, as shown in Fig. 3c. Now, we find a much lower fluctuation intensity, with a maximum of 0.08 for $NH_{3,total}$ compared to 30 for $NH_{3,plume}$. The U-shape in shown in the transect of Fig. 3a is replaced by an approximately Gaussian shape, with the highest fluctuation intensities at the centerline of

the plume. This centerline fI decreases with distance from the source and becomes indistinguishable from the out-of-plume fI after approximately 1 km distance, i.e. a rough estimate for $BD_{fI}$.

Finally, Fig. 3d shows that the emission plume leads to a positive flux (emission) for $NH_{3,total}$ in proximity of the emission source, while the flux is negative (deposition) outside the plume. Note that significant fluctuations are found in the flux over the full domain, with $\sigma_{F,bg}$ = 0.0065 ppb m s$^{-1}$ (prescribed $F_{sfc.}$ = -0.045 ppb m s$^{-1}$) for $NH_{3,bg}$. Similar to fI$_{total}$ in Fig. 3c, the transects for the NH$_3$ flux are approximately gaussian in shape, with the peak values close to the plume centerline at $y/y_{plume}$ = 0. After approximately 1 km at the approximate plume centerline, the in-plume flux becomes visually indistinguishable from the background, i.e. a rough estimate for BD$_F$. This positive anomaly is the result of the emission source being within the footprint of these receptors.

## 3.2  Quantitative analysis of the NH$_3$ emission plume impact

We apply the concept of blending-distance in Fig. 4 to the fluctuation intensity (a), flux (b) and intermittency (c). The markers represent the value at each individual grid point on the 37.5 m horizontal plane, the continuous orange line represents the gridpoint with the highest value within a 50 m moving window (maximum), the orange dotted line represents the plume centerline and the purple dashed and continuous lines represent the blending-distances for their respective threshold.

We interpret the calculation of the blending-distance based on 4 arbitrary threshold levels (5%, 10%, 25% and 50%) for fI and F, shown in Fig. 4a and b. The distance at which the maximum value of PC$_X$ drops below the threshold level is the blending-distance. The sensitivity of BD to these thresholds will be discussed in detail in Sect. 4.1, using Fig. 6 and 7. Additionally, the intermittency in Fig. 4c shows that the emission plume is quantifiable up to over 2.4 km distance.

Starting with the fluctuation intensity (Fig. 4a), PC$_{fI}$ peaks at a relative change of about 300%, caused by the NH$_3$ emission plume. BD$_{fI}$ decreases non-linearly from 0.7 km to 1.9 km with the thresholds decreasing from 50% to 5%. Figure 4b shows that the emission plume has a larger impact on NH$_3$ flux measurements than on the fluctuation intensity of the NH$_3$ molar fraction. The large difference between the emission strenght (45 ppb m s$^{-1}$) and the deposition (-0.045 ppb m s$^{-1}$) result in a maximum PC$_F$ of about 1200% in close proximity of the emission source. The long tail of PC$_F$ indicates that the turbulent fluctuations in the emission plume affect flux measurements over several kilometers. As a result, BD$_F$ increases from 1.2 km to 2.9 km for decreasing thresholds, significantly longer distances then our 1 km qualitative estimate based on Fig. 3d. Note that Fig. 3d shows that the flux changes sign in proximity of the emission source and that this sign change is not reflected in Fig. 4b.

Figure 4 shows that the centerline and the maximum statistics give similar results, indicating that the highest values of PC are found at the plume centerline, though with some variability. These variabilities are visualized in Fig. 5, which shows the spatial structure of the percentage change in grayscale for the fluctuation intensity (PC$_{fI}$) in (a) and the ammonia flux (PC$_F$) in (b). The colored lines in these panels represent the blending-distances for different thresholds. The right panels show these same blending-distances for different angles from the plume centerline (W), representing different wind directions. Fig. 5 shows large variability in the blending-distance, especially for the 5% and 10% threshold levels, as a result of the chaotic nature of turbulence.

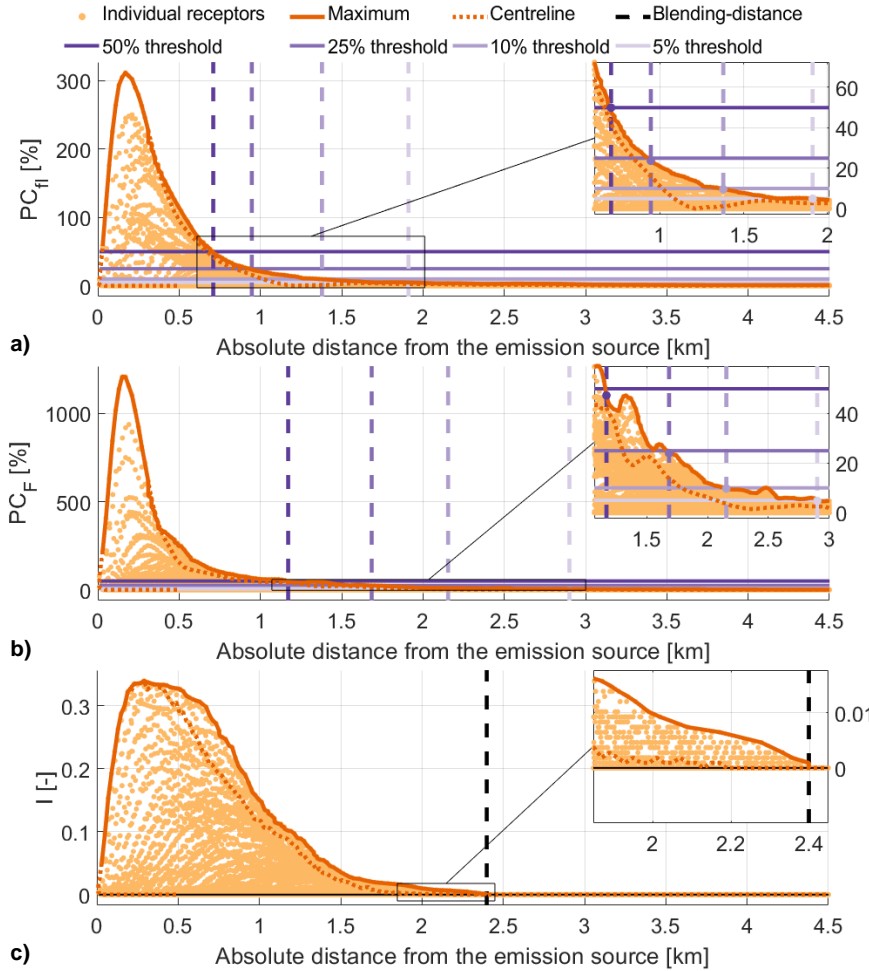

**Figure 4.** The percentage change of $NH_{3,total}$ relative to $NH_{3,bg}$ against absolute distance from the $NH_3$ emission source. The panels show fluctuation intensity (a), $NH_3$ flux (b) and intermittency (c). Highlighted are the maximum value within a 50 m moving window (orange) and the plume centerline (dotted orange). Blending-distances (purple dashed) are calculated based on three thresholds at 5%, 10%, 25% and 50% (purple continuous).

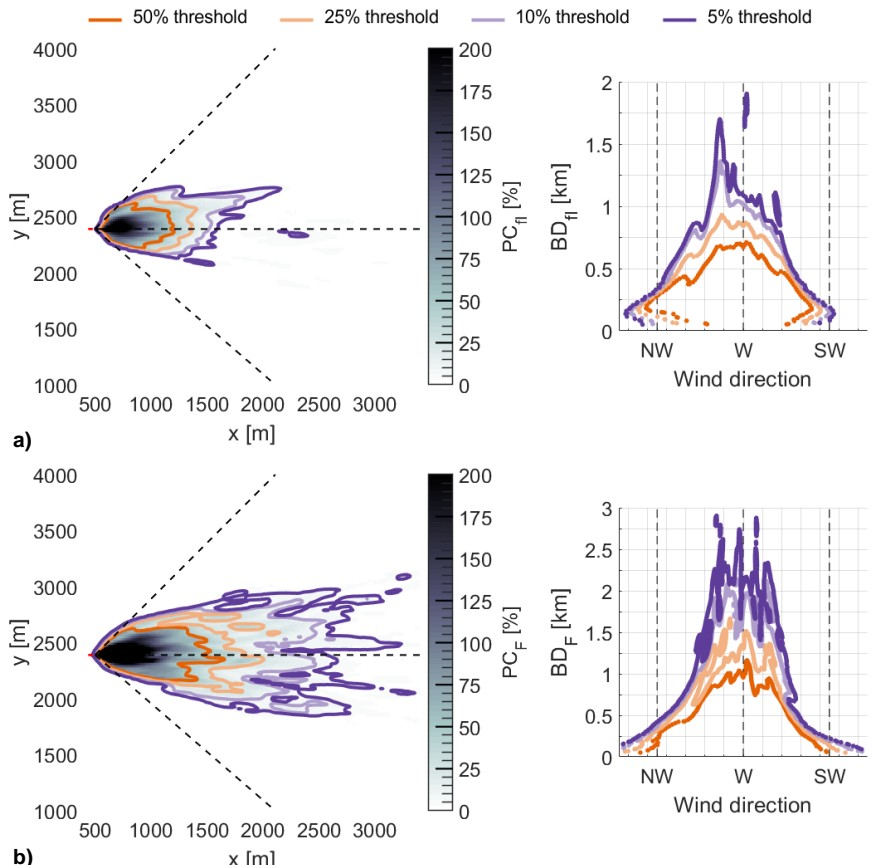

**Figure 5.** The left panels show the spatial structure of the percentage change in grayscale for the fluctuation intensity ($PC_{fI}$) in (a) and the ammonia flux ($PC_F$) in (b). The colored contour lines show the locations where the 50% (orange), 25% (light orange), 10% (light-purple) and 5% (purple) thresholds are met, representing the blending-distance (BD). The right panels show these blending-distances as a function of different angles from the plume centerline (W), with these angles representing the wind direction.

## 4 Discussion

### 4.1 Sensitivity of blending-distance to meteorological and NH$_3$ pollution variables

We study the sensitivities of $BD_{fI}$ and $BD_F$ to a range of meteorological, NH$_3$ pollution parameters and model resolution and simulated measurement height (Table 1). The results of the sensitivity study are shown in Fig. 6 and 7 for an arbitrary set of threshold ranging from 5% (orange dashed) to 50% (orange dotted), representing the maximum acceptable difference in fI and F caused by the emission plume in %.

Starting with $BD_{fI}$, Fig. 6 shows that $BD_{fI}$ ranges roughly between 0.5 and 3.0 km; a first-order estimate of the minimum distance for NH$_3$ molar fraction measurements. There is a negative correlation between $BD_{fI}$ and the choice in threshold, i.e. increasing the threshold level decreases $BD_{fI}$. We generally find that $BD_{fI}$ decreases nonlinearly by approximately 1.0 km

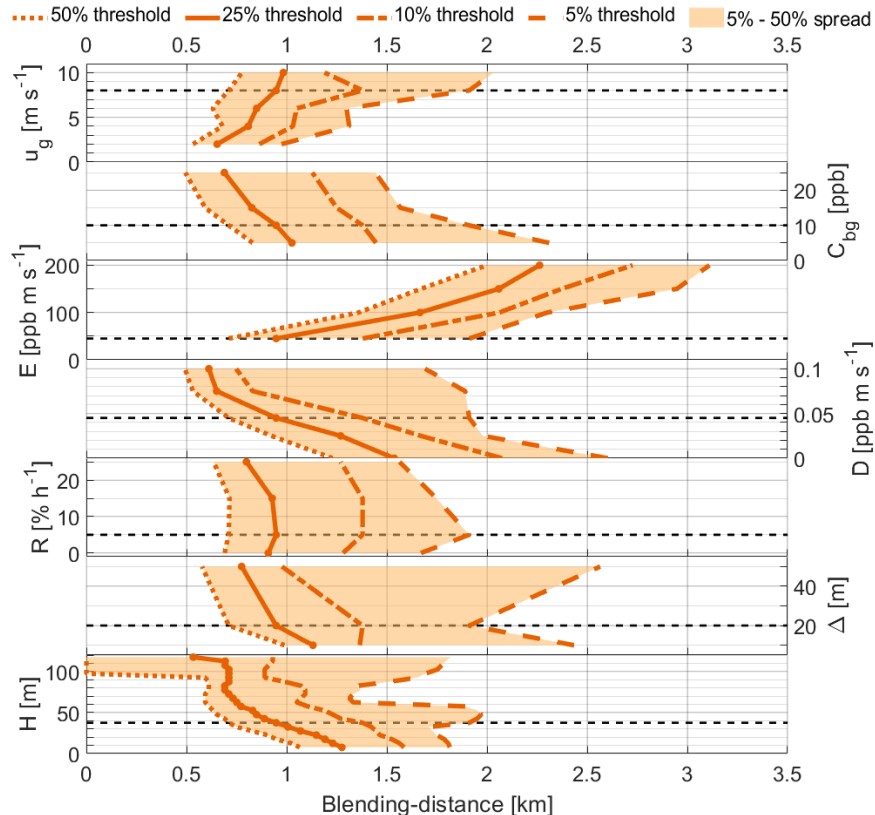

**Figure 6.** The sensitivity of $BD_{fl}$ to the geostrophic wind speed ($u_g$), initial background molar fraction ($C_{bg}$), emission strength (E), deposition strength (D), chemical reaction rate (R), model resolution ($\Delta$) and simulated measurement height (H). $BD_{fl}$ is determined for threshold levels ranging from 5% (orange dashed) to 50% (orange dotted).

when increasing the threshold level from 10% to 50%, halving $BD_{fl}$, highlighted by the large difference between the 10% (dashed-dotted) and 5% (dashed) threshold levels for both $BD_{fl}$ and $BD_F$. We discuss the individual variables of Fig. 6 from

top to bottom, starting at the mesoscale ($u_g$), down to the micrometer scale (R) and finishing with the model resolution and simulated measurement heigth.

The geostrophic wind speed ($u_g$) is one of the main drivers of turbulent mixing and transport of the plume (Dosio et al., 2003; Vrieling and Nieuwstadt, 2003; Dosio and Vilà-Guerau de Arellano, 2006). Figure 6 shows a positive correlation between $BD_{fl}$ and $u_g$. By varying $u_g$ we move from a convection-driven boundary layer ($u_g = 2$ m s$^{-1}$) to more shear-driven meteorological

conditions ($u_g = 10$ m s$^{-1}$). In a convection-driven boundary layer, turbulent mixing is rather weak and the NH$_3$ emission plume rises from the surface as convection plumes are the main drivers of turbulent mixing. Under these conditions, in-plume molar fractions are very high, but horizontal transport of the emission plume is weak, resulting in a low $BD_{fl}$. For shear-driven conditions, the NH$_3$ emission plume tends to stick to the surface as the increased horizontal wind speed enhances horizontal transport and turbulent mixing. The enhanced horizontal transport and emission plume sticking to the surface should

significantly increase $BD_{fI}$, but the enhanced turbulent mixing counteracts these processes by reducing the $NH_{3,plume}$ molar fraction and fluctuations. This is shown in Fig. 6 and explains why the sensitivity of $BD_{fI}$ increases for lower threshold levels (5%), as smaller plume fluctuations will reach long distances in shear-driven conditions.

One panel below, Fig. 6 shows a negative correlation between $BD_{fI}$ and the initial background molar fraction ($C_{bg}$), i.e. the regional level of $NH_3$ pollution. The first cause of the negative correlation is the higher average molar fraction, which lowers

relative weight of the $NH_{3,plume}$ fluctuations ($\sigma_{plume}$) when $fI_{total}$ is calculated following Eq. 2. Additionally, increasing $NH_{3,bg}$ leads to a large difference in the $NH_3$ air mass characteristics at the top of the boundary-layer. The exchange between the boundary layer and free tropospheric air masses throught entrainment increases $\sigma_{bg}$ at 37.5 m from 0.13 ppb ($C_{bg}$ = 5 ppb) to 0.38 ppb ($C_{bg}$ = 25 ppb), resulting in an increased $fI_{bg}$. Both processes reduce the magnitude of $PC_{fI}$ with increasing $C_{bg}$, reducing $BD_{fI}$.

At the local scale, Fig. 6 shows a clear positive and negative correlation when varying emission strength (E) and deposition strength (D) respectively. Both variables directly affect one of the main drivers of turblent mixing: heterogeneity. Increasing the $NH_3$ emission strength of the local (heterogeneous) source directly increases $fI_{plume}$, increasing $BD_{fI}$. Varying the deposition on the other hand, directly affects the vertical gradient of the $NH_3$ molar fraction near the surface, increasing $fI_{bg}$ for increasing D and therefore reducing $BD_{fI}$.

We only briefly touch upon the chemical conversion rate (R), as Fig. 6 shows that varying R does not significantly affect $BD_{fI}$. R is applied uniformly to the 3D domain and has little effect on turbulent mixing. Note that our simplified representation of chemistry could lead to a potential underestimation of the impact of chemistry on $BD_{fI}$, as our approach is unable to resolve potential non-linear effects of turbulent mixing on the in-plume chemical reaction rate near the emission source (see discussion in Sect. 4.2).

Next, we vary the model resolution ($\Delta$) in Fig. 6 and find that $BD_{fI}$ is weakly sensitive to the model resolution. The results indicate that the calculation of the blending-distance does benefit by increasing the simulation resolution. However, there is a trade-off between the computational costs of the simulation and the resolution.

Finally, Fig. 6 shows two regimes in the sensitivity of $BD_{fI}$ to the simulated measurement height (H). For the 50% threshold, BD decreases by about 500 m with height up to 90 m. Above 90 m, there is a transition where $BD_{fI}$ rapidly goes to zero. In this

second regime, the simulated measurements are located above the plume centerline. From there on, $fI_{plume}$ rapidly decreases with height until $PC_{fI}$ does not reach the 50% threshold and $BD_{fI}$ becomes zero. This rapid decrease is a result of the simulated measurements being located above the emission plume, as the height of plume does not reach above 150 m for the first 1.5 km horizontal distance. The height of this transition increases with decreasing threshold levels as the thresholds become more sensitive to smaller $NH_{3,plume}$ fluctuations.

Figure 7 shows the results of the sensitivity study for $BD_F$ (Table 1). Both the blending-distance for molar fraction measurements ($BD_{fI}$) and for flux measurements ($BD_F$) can be interpreted as a inverse footprint analysis, as we estimate the area affected by the emission source. The results of the sensitivity study of $BD_F$ however, are different from the $BD_{fI}$ results, as the footprints for flux and molar fraction measurements are not the same. Footprint for flux measurements are smaller than those of molar fraction measurements (Rannik et al., 2000; Kljun et al., 2003; Vesala et al., 2008). However, comparing $BD_{fI}$ to the

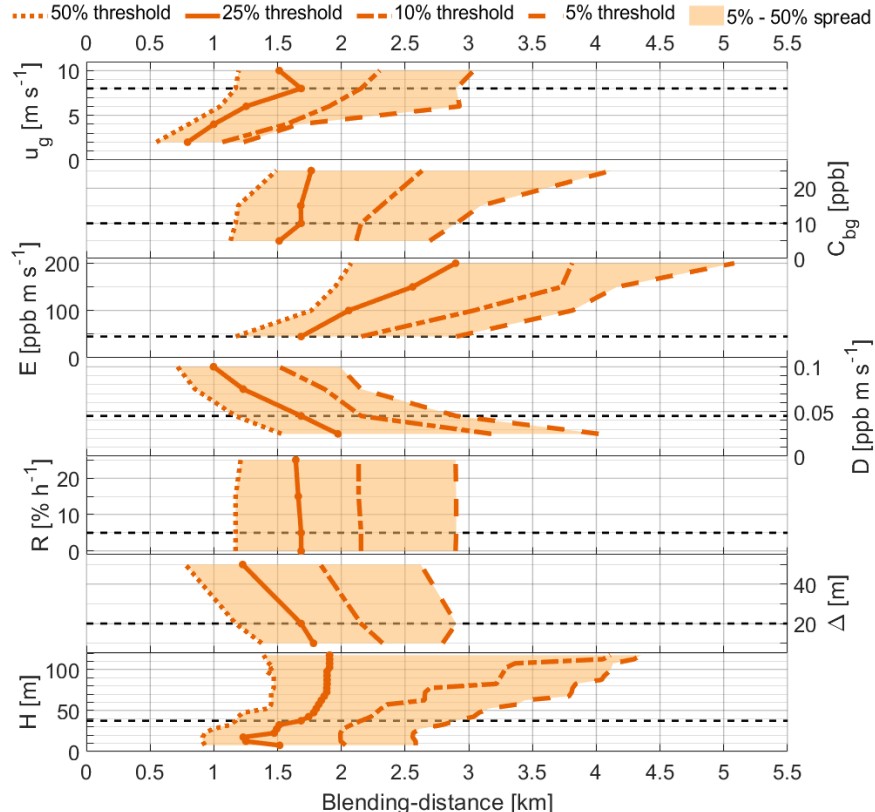

**Figure 7.** The sensitivity of $BD_F$ to the geostrophic wind speed ($u_g$), initial background molar fraction ($C_{bg}$), emission strength (E), deposition strength (D), chemical reaction rate (R), model resolution ($\Delta$) and simulated measurement height (H). $BD_F$ is determined for threshold levels ranging from 5% (orange dashed) to 50% (orange dotted).

footprint of $NH_3$ molar fraction measurements is not straightforward, as $BD_{fl}$ is based on the $NH_3$ fluctuation intensity, not the molar fraction. It is therefore interesting to determine whether the results of the sensitivity study of $BD_F$ will differ compared to the results of $BD_{fl}$.

     When analyzing Fig. 7, we find that there are indeed differences between $BD_F$ and $BD_{fl}$. $BD_F$ is significantly longer, ranging from 0.75 to roughly 5 km, indicating that $NH_3$ flux measurements are more sensitive to the emission plume. Note that we

removed the results for D = 0 ppb m s$^{-1}$. Here, $F_{bg}$ approaches zero, resulting in infinitely large $PC_F$ and unrealistic $BD_F$ values, following Eq. 6.

     One of main differences between $BD_F$ and $BD_{fl}$ are found in the sensitivity to the threshold levels (5% to 50%). $BD_F$ is more sensitive to the different threshold levels compared to $BD_{fl}$. This is in agreement with the results shown in Fig. 4b, where we discussed that $PC_F$ is significantly larger than $PC_{fl}$, with a longer tail. As a result, the non-linear effect of the aforementioned

long tail in $PC_F$ (Fig. 4b) increases $BD_F$ for low threshold levels. Despite these differences, the same arbitrary set of thresholds are used for both $BD_{fl}$ and $BD_F$.

Significant differences between $BD_F$ and $BD_{fl}$ are also found in the sensitivity to the geostrophic wind speed ($u_g$) and the simulated measurement height (H). Both variables directly affect the footprint of the simulated flux measurements. In shear-driven turbulent conditions (high $u_g$), the footprint of the measurement is elongated compared to convective conditions. This reduces the width of the footprint and lengthens the up-wind distance at which the emission source can be measurement, thus increasing $BD_F$. Increasing H also increases the footprint of the measurements, but there is no elongation of the footprint. As a result, $BD_F$ has a strong positive correlation to $u_g$ but is only weakly correlated to H, except for the lower threshold levels.

Fig. 7 appears to show that $BD_F$ has a weak positive correlation with increasing $C_{bg}$. This is mainly attributed to an increase in the spatial variations of the background $NH_3$ flux, which increases from $\sigma_{F,bg} = 0.0065$ ppb m s$^{-1}$ for $C_{bg} = 10$ ppb (Fig. 3d) to $\sigma_{F,bg} = 0.015$ for $C_{bg} = 25$ ppb. As a result, the fluctuations in $PC_F$ shown in Fig. 4b increase in amplitude and frequency which particularly affects the low threshold levels of 5% and 10% .

There are also strong similarities between the sensitivity of $BD_F$ and $BD_{fl}$. Both Fig. 6 and 7 show that the blending-distance is only weakly sensitive to the chemical reaction rate (R) and the model resolution ($\Delta$). For both molar fraction and flux measurements, the emission strength (E), deposition (D) and to a lesser extent the geostrophic wind speed ($u_3$ are the driving variables of the blending-distance.

## 4.2   Uncertainty of the blending-distance estimation

The turbulent dispersion of the emission plume is chaotic by nature and driven by a wide range of factors. We therefore carry out a systematic analysis on how these factors, as well as the model resolution, influence the the relationships between emission and the simulated in-field measurements. The chaotic nature of turbulence results in random variations in both the emitted $NH_3$ (Fig. 3a and b) and the background $NH_3$ (Fig. 3c and d). These random fluctuations lead to variability in the calculation of the blending-distances, leading to uncertainty in the blending-distances presented in this study. The variability increases when using the lower threshold levels (e.g. 5% and 10%), as is visualized and discussed in Sect. 3.1 and 3.2. The variability could be reduced by increasing the length of the analysis window, i.e. increasing the averaging time to filter out the small and short spatio-temporal turbulence variability.

Increasing the length analysis window however, means that the blending-distance is calculated using a wider range of boundary layer dynamics and variations in the thermodynamic variables. Boundary layer dynamics are especially relevant in the morning and early afternoon, when the boundary-layer grows and air from the residual layer and free troposphere is entrained, or in the afternoon when turbulence decays (Pino et al., 2006). It leads to entrainment being one of the dominant processes driving the $NH_3$ diurnal variability (Wichink Kruit et al., 2007; Schulte et al., 2021). We show in Fig. 2 it leads leads to large fluctuations in $NH_{3,\,bg}$, significantly increasing $fl_{bg}$. We therefore filter out the impact of boundary layer dynamics and variations in the thermodynamic variables with our choice of analysis window from 14:00 and 17:00 CEST, in order to find a first-order estimate of the blending-distance. We do recommend a follow-up study on the role of boundary dynamics.

Finally, there is a downside to of our simplified representation of chemical transformations, in that it is applied uniformly to the 3D domain. In reality, the equilibrium molar fractions for these chemical transformations are related to temperature and humidity and results in a near-surface $NH_3$ gradient of the $NH_3$ molar fraction (aan de Brugh et al., 2013). Therefore, we are

likely to underestimate the role of chemical transformations and overestimate $BD_{fI}$, as turbulent mixing of this near-surface gradient increases $fI_{bg}$.

The blending-distance cannot be captured by a single number. This is partly due to the uncertainty involved in calculating the blending-distance, but the blending-distances is most of all an integrated variable. Several processes are captured by the blending-distance in one single variable, including the chaotic nature of turbulent plume dispersion, convective and shear induced turbulence, atmospheric pollution levels and surface heterogeneity. As shown in Sect. 4.1, each of these processes impacts the blending-distance differently. Despite its complexity, the blending-distance is a useful variable since it is an integrated variable; all the aforementioned processes are represents in this distance at which the impact of an emission plume is negligible with respect to the background.

The applicability of the results presented here depend not only on the meteorological and $NH_3$ pollution factors, but also on the physical context of the measurement site. This study is based on the Ruisdael CESAR Observatory at Cabauw, which is located on flat agricultural grassland with surface elevation changing only by a few meters over 20 km. A different physical context, like a heterogeneous surface which changes the turbulent properties (Ouwersloot et al., 2011), is likely to significantly affect the resulting blending-distances. With the simulation framework presented here, the blending-distance can be calculated for specific weather conditions and for the physical context of the measurement site, providing a more accurate assessment of the impact of nearby emissions on $NH_3$ observations at a specific measurement site. The results presented in this study provide a valuable first estimate of, and discussion on, a typical blending-distance and its driving variables.

### 4.3 Blending-distance literature for passive tracers

Evaluating the blending-distance results against typical literature on plume dispersion is a difficult exercise. The topic is generally not mentioned as these studies focus on the release of passive scalars in an unpolluted environment and only few studies even research (near) surface releases (Cassiani et al., 2020). Normalization of both distance from the source and the in-plume molar fraction further complicates the interpretation of literature results.

We therefore try to estimate the order of magnitude of the blending-distance based on the in-plume molar fraction and fluctuation intensity of plume dispersion modeling studies. Following figures by Dosio et al. (2003) and Dosio and Vilà-Guerau de Arellano (2006), we find that the in-plume molar fraction rapidly decreases for a convection-driven boundary layer ($-z/L \geq 40$ and $u_*/w_* \leq 0.2$) at the surface up to roughly 6 km distance, after which it starts to level off. Similar results are found for the fluctuation intensity, although the results are less pronounced for the near-surface release experiments. The 6 km distance approximately doubles for shear-driven boundary layers ($-z/L \sim 40$ and $u_*/w_* \sim 0.46$). The observations shown in Figure 10 by Mylne and Mason (1991) show that the observed fluctuation intensity also decreases with distance, but levels after roughly 15 km distance from the emission source. We use these distances at which the plume statistics start to level off as an estimate of the order of magnitude of the blending-distance, indicating that the blending-distance could be in the order of several kilometers (6 to 15 km), based on plume dispersion literature.

These rough estimates of 6 to 15 km distance are significantly larger than the blending-distances presented in this study. Such long distances between source and measurement site would not make feasible requirements in densely agricultural regions, but

are likely an overestimation of the blending-distance. These estimates are based on the molar fraction and fI of the emission plume, with no representation of background ammonia levels. The latter is especially important, as we show in Sect. 3.1 and 3.2 that the impact of the emission plume rapidly decreases relative to the turbulent background ammonia, while the emission plume itself can be detected for several kilometers as indicated by the intermittency.

### 4.4 Blending-distance literature for ammonia measurements

Articles on ammonia measurements in close proximity of an emission source implicitly include all relevant processes. Such studies could also provide a qualitative, perhaps more realistic, evaluation of the $NH_3$ blending-distance results presented here. In-field measurements show that the $NH_3$ molar fraction exponentially decreases with distance from the source, with measurements close to the background molar fraction after 300 to 500 m (Fowler et al., 1998; Sommer et al., 2009; Shen et al., 2016). Similar results were obtained in an intercomparison study of short-range atmospheric dispersion models by Theobald

et al. (2012), at horizontal resolutions of 25 - 50 m and receptors at 100 m intervals along four radial directions (N, E, S and W). However, such measurements are typically arranged in a few lines downwind of the source, with only a handful of measurements over a distance of 300 to 1000 m. At these short distances, plume dispersion is dominated by meandering of the plume (Nieuwstadt, 1992) and the in-plume molar fraction measurements are underestimated as a result, especially given the averaging times of these measurements ranging from several hours up to multiple weeks.

Finally, we can evaluate our findings against measurement site requirements of air quality networks. The Dutch air quality network and the EMEP (European Monitoring and Evaluation Programme) network do set requirements on the minimum distance from emission sources, no references to scientific studies are provided. Back in 1990, the Dutch network required a minimum distance for $NH_3$ sites of 300 - 500 m from $NH_3$ point or area sources, depending on source strength (Boermans and Erisman, 1990). This is in line with the literature on measurements in proximity of emission sources discussed earlier, but

closer than the blending-distances presented here. Currently, no hard requirements are in place in the Netherlands, although the potential impact of $NH_3$ sources is still recognized (Wichink Kruit et al., 2021). At a European level, EMEP measurement sites require a 2 km minimum distance for measurements nearby stabling of animals and manure application, depending on the number of animals and field size (Schaug, 1988; EMEP/CCC, 2001). This 2 km distance is in line with our recommendations, although the results in this study indicate that distances below 2 km could also be sufficient.

### 4.5 Towards an $NH_3$ virtual testbed: integrating fine-scale simulations with advanced observations

This study is the first which specifically addresses the regional representativity of ammonia measurements in proximity of an emission source. The systematic analysis presented in Fig. 6 and 7 can be used as a reference when interpreting in-field $NH_3$ measurements. Additionally, the simulation framework can be applied to individual locations and study the representativity of (potential new) measurement sites under the local conditions, using the concept of blending-distance. One can expanded the

470 simulation framework to include multiple sources, area sources, each with an unique passive scalar, as well as heterogeneous surface conditions (Ouwersloot et al., 2011), to simulate the local $NH_3$ conditions.

The DALES model has proven to be flexible, allowing for simulations of a convective, sheared convective, stable and cloud topped boundary layer (Verzijlbergh et al., 2009; Heus et al., 2010). The fine-scale simulation framework will be included in the Ruisdael CESAR Observatory at Cabauw (https://ruisdael-observatory.nl), a nationwide observatory for measurements and modeling of the atmosphere and air quality. It can be a powerful tool in future ammonia research, e.g. in preparation of (emission) measurement campaigns or to improve interpretation of $NH_3$ (flux) measurements. Furthermore, we want to stress that the methods presented here are not limited to ammonia, but can be used for any gas for which the relevant processes occur at high spatio-temporal resolution.

We recommend to expand the simulation framework to create a testbed to study $NH_3$ at high spatio-temporal resolution, including all processes relevant to the $NH_3$ diurnal variability. The main additions should be a dynamic parameterization of the surface-atmosphere exchange, e.g. DEPAC (van Zanten et al., 2010), and a thermodynamic chemistry module, e.g. ISORROPIA version 2 (Fountoukis and Nenes, 2007). With these additions, on top of the existing possibility to distinguish between background and emitted $NH_3$, the fine-scale simulation framework with explicitly resolved turbulence will be well suited to study short-range dispersion of ammonia, e.g. deposition in close proximity to emission sources and the impact of turbulent micromixing on the chemical reaction rate. Such studies are typically performed using models where turbulence is parameterized or using Gaussian plume models (Loubet et al., 2006; Sommer et al., 2009; van der Swaluw et al., 2017). Furthermore, the addition of a thermodynamic chemistry module can lead to new insights on $NH_3$ flux measurements. The equilibrium molar fractions of the $NH_3$ gas-aerosol transformations depend on the atmospheric temperature and humidity, resulting in a near-surface molar fraction gradient. This gradient leads to an underestimation of the $NH_3$ deposition flux of about 0.02 $\mu$g m$^{-2}$s$^{-1}$ when using the flux-gradient method (Nemitz et al., 2004). With these additions to the simulation framework, the virutal $NH_3$ testbed can be used improve the interpretation of $NH_3$ flux measurements.

## 5 Conclusions

This paper presents a fine-scale simulation framework with which we assess the regional representativity of $NH_3$ molar fraction and flux measurements in proximity of a typical $NH_3$ emission source. We aim to translate concepts from the fields of plume dispersion and fine-scale simulations to support the analysis of $NH_3$ observations in areas characterized by $NH_3$ (point) source emissions, including realistic representations of $NH_3$ surface-atmosphere exchange and chemical gas-aerosol transformations. The concept of a blending-distance is introduced to systematically analyze the impact of the emitted $NH_3$ on simulated measurements, relative to a background concentration. Following this approach, we define a first-order estimate of a minimum distance requirement between regional representative measurements and a typical $NH_3$ emission source.

By means of fine-scale simulation of atmospheric $NH_3$, we investigate the representativity of $NH_3$ measurements from kilometer to meter scales in proximity of a typical emission source. The fine-scale simulation framework presented has proven to be a powerful and flexible tool for future research on ammonia, or any gas for which the relevant processes occur at high spatio-temporal resolution. The simulation framework with explicitly resolved turbulence not only enables us to quantify the variability in $NH_3$ measurements, but also to analyze and quantify the individual contribution of the $NH_3$ emission plume.

The concept of blending-distance presents a consistent criterion, based on second order statistics, for the minimum distance at which the impact of the emitted $NH_3$ is estimated to be indistinguishable from the variability of the background $NH_3$. Following this approach, we perform several numerical experiments to analyze the sensitivity of the blending-distance to a variety of meteorological and $NH_3$ pollution variables, centered around the flat grassland at Ruisdael CESAR Observatory at Cabauw. This systematic analysis shows a strong sensitivity to the emission strength, deposition and the threshold level used in the calculation, and to the stability of the (convective or shear dominated) boundary layer. Furthermore, we find that the blending-distances differ for $NH_3$ molar fraction and flux measurements, with flux measurements being more sensitive to the $NH_3$ emission plume. Following this sensitivity analysis, we conclude that $NH_3$ measurements at the CESAR Observatory should be taken at a minimum distance of 0.5 - 3.0 km or 0.75 - 4.5 km distance from an emission source, for measurements of the $NH_3$ molar fraction or flux respectively.

*Code availability.* The modified version of the Dutch Atmospheric Large-Eddy Simulation (DALES) version 4.2, including data processing scripts and documentation, is available at http://doi.org/10.4121/19869478.

*Author contributions.* RS and JV worked on the conceptualization and developed the methodology of the numerical experiments. RS and BS made modifications to the software, i.e. the DALES model. The numerical experiments were performed by RS, who also analyzed and visualized resulting data. The manuscript draft is written by RS and reviewed by both JV and MZ. Finally, the project was supervised by JV and MZ.

*Competing interests.* The authors declare that they have no conflict of interest.

*Acknowledgements.* This research was financed by the Dutch National Institute of Public Health and the Environment (RIVM) within the framework of the Project 36.7: Monitoring of dry ammonia deposition, which is performed by order, and for the account, of the Dutch Ministry of Agriculture, Nature and Food Quality. The numerical simulations were performed with the supercomputer facilities at SURFsara and financially sponsored by the Netherlands Organization for Scientific Research (NWO) Physical Science Division (project number 2021/ENW/01081379).

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
