# Peer review of "Assessing representativity of NH3 measurements influenced by boundary-layer dynamics and turbulent dispersion of a nearby emission source"

_Atmospheric Chemistry and Physics, 2021_

## Author Response (AR2)

**Author's response**

**General response**

We thank both Referee #1 and Referee #2 for their thorough reading and for her/his comments. The comments were very constructive and have triggered us to extend our analysis based on additional numerical simulations. We have taken the majority of the comments into account. However, we felt that some of the results were outside the scope of this study, but we are discussing them in this document. Below, we have provided a point-to-point response to each comment made by the individual Referees. Note that we use Roman numerals for the Figures in this document, to avoid confusion when we refer to Figures in the revised manuscript.

We apologize for the delayed response. In November 2021, the Cartesius supercomputer we used to perform the DALES experiments was replaced by a new Dutch National Supercomputer called Snellius. This caused several problems compiling the model which needed to be resolved and there were several moments of maintenance delaying our experiments. We also noticed that the model results on the new supercomputer were slightly different (within numerical rounding error) from those presented in the original manuscript, but the conclusions presented before still hold for the new results. For the sake of consistency and quality, we have decided to redo all experiments and updated the Figures for the revised manuscript. Finally, we have decided to add the blending-distance results for the 10% threshold level to the Figures, as it nicely highlights the non-linear relation between the blending-distance and the (lower) threshold levels.

**Response to Referee #1**

1. **It would be interesting discuss a little bit further the representativeness of the conclusions achieved in a different orographic context, the variability throughout 24 hours due to atmospheric stability changes and the potential effect of this variables in blending-distance variability**

The Referee raises several interesting points related to changes in the turbulent and atmospheric stability conditions. In 24 hours, the atmospheric stability ranges from stable/neutral conditions at night to convective conditions in the afternoon. The orography of a region or land use contrast can induce the arrival of an air mass characterized by different thermodynamic characteristics, resulting in a change in the turbulent and stability conditions in the boundary layer. In both cases, the changes in turbulent properties could affect the turbulent mixing of the emission plume and affect our estimates of the blending-distance.

In the manuscript, we actively filter out these processes with our choice of the analysis window (14:00 – 17:00 CEST) in order to give a first-order estimate of the blending-distance. However we do agree with the Referee that it would be interesting to further discuss this topic. Following her/his advice, we perform 3 new experiments. Firstly, we perform an experiment where we release the $NH_3$ emission plume, activate the surface deposition and activate the chemical conversion rate at 8:00 CEST and continue until 17:00 CEST, the full-day experiment. With this experiment, we study the sensitivity of the blending-distance to the diurnal variability. Secondly, we perform 2 experiments where we study the sensitivity to a change in turbulent properties by simulating the arrival of a mesoscale phenomenon, i.e. a sea breeze front. We decided to simulate a sea breeze, because the arrival of a sea breeze front is well documented at the CESAR Observatory site (Arrillaga et al., 2018) and it is a similar phenomenon as the orography can induce. Note the flat conditions around the 213-meter tower of the CESAR Observatory at Cabauw.

The full-day experiment

The full-day experiment is an adaptation of the reference experiment as shown in Table 1 of the manuscript. The only change with respect to the reference experiment is that the emission, the

deposition and chemical conversion rate are activated at 8:00 CEST instead of at 12:30 CEST As a result, a lower $NH_3$ molar fraction are found at 14:00 CEST compared to the experiments presented in the revised manuscript; about 8 ppb compared to the original 10 ppb. We vary the start and end time of the analysis window from 8:00 to 17:00 CEST, while maintaining a length of three hours. The resulting blending-distance for the fluctuation intensity ($BD_{fl}$) is shown in Figure **_Ia_** below, for each of the analysis windows between 8:00 and 17:00 CEST. To quantify the turbulent conditions of the boundary layer, we also show the turbulent kinetic energy (TKE), integrated over the boundary layer height at the corresponding time intervals, in Figure **_Ib_**.

[Figure]

*Figure I. $BD_{fl}$ for each 3-hour long analysis window (a) and the corresponding horizontal average TKE integrated over the convective boundary layer (CBL) height.*

Following Figure **_I_**, we find a clear relation between the blending-distance and the turbulent kinetic energy. During the morning transition, there is little turbulent mixing and $BD_{fl}$ reaches up to several kilometers for each threshold level. As the TKE increases, the emission plume disperses more rapidly and $BD_{fl}$ decreases. $BD_{fl}$ starts to increase again, as entrainment no longer plays a role after 13:30 CEST (Fig. 2 in the manuscript).

The blending-distance of the $NH_3$ flux ($BD_F$) is shown in Figure **_IIa_** below, also accompanied by the TKE integrated over the CBL height in Figure **_IIb_**. Note that $BD_F$ for the 5% threshold is not shown within the figure limits for the 8:00 – 11:00 CEST analysis window as it reaches up to 9 km, while the 10% threshold level only reaches 2.3 km. Figure **_IIa_** shows that $BD_F$ increases with increasing TKE, except in the late afternoon.

[Figure]

*Figure II. $BD_F$ for each 3-hour long analysis window (a) and the corresponding horizontal average TKE integrated over the convective boundary layer (CBL) height.*

The changes in $BD_F$ appear to be strongly driven by entrainment and the specific configuration of the boundary layer structure. Figure **_IIIa_** shows that an overshoot of the boundary layer height at about 10:30 CEST. However, Figure **_IIIb_** shows that the turbulent mixing of the boundary layer with the $NH_3$ residual layer and free troposphere takes considerably more time and happens between roughly 11:00 and 13:00 CEST. This is clearly visualized by the strong negative flux at 12:00 CEST in Figure **_IIIc_** and can be also be inferred from the sudden decrease in $NH_{3, bg}$ in Figure **_IIIb_**. As shown in Figure **_II_**, $BD_F$ increases when data between 12:00 and 13:00 CEST are included in the analysis period, which is the time period where entrainment is strongest.

[Figure]

*Figure III. The boundary layer height evolution (a), the vertical profiles of the $NH_{3, bg}$ molar fraction (b) and the vertical profiles of $F_{bg}$ between 10:00 and 14:00 CEST.*

The Sea breeze experiments.

The sea breeze experiment is designed based on over 100 days of sea breeze observations at the CESAR Observatory at Cabauw, described by Arrillaga et al., (2018). As mentioned, this experiment is conducted to study the impact on the $NH_3$ turbulent mixing by any mesoscale circulation disturbance that generates advection of heat and moisture. As such, the sea breeze is represented by adding a large scale forcing term to the model for the potential temperature and specific humidity. At the starting time of the large scale forcing, an additional source or sink is added to the model up to a height of 1700 m (just below the boundary layer height). We perform 2 sea breeze experiments; a realistic sea breeze based on the measurements of Arrillaga et al., (2018) (SB) and a more extreme sea-breeze experiment (SB$_{extreme}$), shown in the table below. In the SB experiment, the sea breeze arrives at 15:00 CEST, or one hour after the start of the analysis phase of the blending-distance, with the changes in θ and $q_t$ based on Figure 8 by Arrillaga et al., (2018). The SB$_{extreme}$ case was performed as well as the effect of the original sea breeze on the boundary layer appeared to be not as strong.

*Table I. The arrival time of the sea breeze the forcing of the potential temperature (θ) and specific humidity ($q_t$), for both the sea breeze (SB) and the extreme sea breeze (SB$_{extreme}$) experiment.*

|  | Sea breeze arrival | dθ/dt | dq/dt |
|---|---|---|---|
| **SB** | 13:00 CEST | -0.75 K h$^{-1}$ | 0.45 g kg$^{-1}$ h$^{-1}$ |
| **SB$_{extreme}$** | 15:00 CEST | -1.50 K h$^{-1}$ | 0.45 g kg$^{-1}$ h$^{-1}$ |

Figure **IV** shows the changes in the temperature and humidity as the sea breeze arrives at 15:00 and 13:00 CEST for SB and SB$_{extreme}$ respectively. For the SB experiment, the cooling by the sea breeze is rather small, reducing the potential temperature by about 1 K at 17:00 CEST. Additionally, the boundary layer inversion weakens as a result of the large scale forcing. The specific humidity increases by over 1 g/kg at 17:00 CEST compared to the reference experiment and a weakening of the inversion is also visible in the humidity vertical profiles. These effects are amplified for SB$_{extreme}$, but in this experiment, an increase of the boundary layer height is observed between 16:00 and 17:00 CEST as well.

[Figure]

*Figure IV. the vertical profiles at each hour from 12:00 to 17:00 CEST of the potential temperature (a) and the total water specific humidity (b) for the reference experiment (solid), the sea breeze experiment (dotted) and the extreme sea breeze experiment (dashed).*

The surface fluxes respond to the changes in the potential temperature and humidity, as seen in Figure **Va**. The latent heat flux (LvE) decreases due to the increase in atmospheric humidity. In response, the latent heat flux (H) increases as the available energy, the net radiation (Q$_{net}$) is not affected. This leads to a change in the evaporation fraction, shown in Figure **Vb**, which shows that a smaller fraction of the available energy is used for evaporation in the late afternoon for the SB experiment, when compared to the reference. In the case of SB$_{extreme}$, these effect is amplified and they start much earlier as the large scale forcing starts at 13:00 CEST. This leads to a large change in the surface fluxes and a larger fraction of the available energy going to the sensible heat flux in the afternoon. This even leads to an increase in the boundary layer height, as shown in Figure **Vc**.

[Figure]

*Figure V. Time series of the net radiation and surface fluxes (a), the evaporation fraction (b) and the convective boundary layer height (c) for the reference experiment (solid), the sea breeze experiment (dotted) and the extreme sea breeze experiment (dashed).*

Focusing on our main interest, how a large-scale disturbance might affect the dispersion of ammonia, we find in Figure **VII** that only little changes in the blending-distances as a result of the sea breeze. Both $BD_{fI}$ and $BD_F$, slightly increase for the SB experiment, but not for the $SB_{extreme}$ experiment. This could be attributed to the changes in the evaporation fraction, as an increased sensible heat flux leads to stronger convection and more vertical mixing of the emission plume. This could reduce the blending-distance near the surface. Still, the effect of the sea breeze on the blending distance is only small.

[Figure]

*Figure VI. The blending-distance based on the fluctuation intensity (a) and based on the NH₃ flux (b) for the reference experiment (Ref.), the sea breeze experiment (SB) and the extreme sea breeze experiment (SBextreme).*

Conclusions

The results of the full-day and the sea breeze (representative of the impact of mesoscale circulations) experiments show that boundary layer dynamics can have a large impact on the blending-distance. The most important process appears to be the mixing of air masses with different ammonia properties, as is the case with entrainment of free tropospheric air. Under convective conditions, changes in the turbulent properties in the boundary layer appear to have less of an impact on both $BD_{fl}$ and $BD_F$. A follow-up study into the role of the stability conditions on the blending distance would be interesting. However, we do believe that it will be necessary to do a more thorough analysis to properly describe the feedback and effects, which is outside the scope of this manuscript. We do plan to continue this research in another chapter of my PhD thesis.

These new results do further stress that there the concept of the blending distance cannot be captured by a single number, as stated in the discussion on "Uncertainty on the blending-distance estimation". This in part due to the chaotic nature of turbulence and the non-linear relations involved in the NH₃ dispersion. The advantage of defining and using a blending-distance, is that it captures (and is affected by) the essential processes that govern the evolution and distribution of atmospheric NH₃. Such processes include the emission strength, convective/shear induced turbulent dispersion, surface heterogeneity and the effect of large-scale forcing on turbulent dispersion. Despite its complexity, the concept is still useful as it integrates in a single value all the above mentioned processes. In this study, we provide a valuable first-order estimate of the distance at which the effect of nearby emissions on observations is negligible and the concepts presented here can be used to study specific meteorological or surface conditions as well.

Changes made to the revised manuscript

Following the results to this comment by Referee #1, we updated the Section 4.2, "Uncertainty on the blending-distance estimation", to include a short discussion on the blending-distance as an integrated variable, starting at line 407.

References

Arrillaga, J.A., Vila-Guerau de Arellano, J., Bosveld, F., Klein Baltink, H., Yagüe, C., Sastre, M. and Román-Cascón, C.: Impacts of afternoon and evening sea-breeze fronts on local turbulence, and on CO₂ and radon-222 transport, Quarterly Journal of the Royal Meteorological Society, 114, 990-1011, https://doi.org/10.1002/qj.3252, 2018.

2. **This study is limited to 3 hours with 30 minute NH₃ flux input (6 values) during the central hours of the day which seems a little bit restricted. It would be interesting for future works to extend this analysis period.**

The Referee makes a fair point on the limitations of the blending-distance calculations for the $NH_3$ flux. While 10 second resolution data is used to calculate the flux, we indeed calculate the $NH_3$ flux every 30 minutes, which results in the blending-distance being calculated with only 6 values. We agree with the Referee that a longer analysis window is desirable and we plan to study the role of boundary layer dynamics on the blending-distance in another chapter of my PhD thesis, as mentioned in response to comment nr 1 by Referee #1. Furthermore, we believe that there are also significant downsides to a longer analysis window with respect to this study specifically.

The analysis window from 14:00 to 17:00 CEST was chosen for several reasons. Firstly, we focus on the afternoon when the boundary layer has fully developed, as we aim to minimize the impact of boundary layer dynamics such as entrainment. Secondly, we start the analysis window at 14:00, as we prefer that the calculation of $BD_F$ is consistent with the calculation of $BD_{fl}$, which requires a 1 hour leading moving average. Finally, the sensitivity study is performed by changing 1 variable, while keeping the other variables constant to break down the complexity of the problem and unravel which are the dominant processes. This is challenging, as changes in variables like the deposition flux or the chemical conversion rate will also affect the background molar fraction. With an analysis window of only 3 hours, this effect is expected to be small, but it could play an important role in the sensitivity study when using longer analysis windows.

Changes made to the revised manuscript
In response to this comment, we updated Sect. 4.2, "Uncertainty on the blending-distance estimation", starting at line 388. Here, we discuss that the variability in $BD_F$ for the lower threshold levels can be reduced with a longer analysis window, but that there is a trade-off in chosing the analysis window, as we aim to minimize the impact of boundary layer dynamics such as entrainment.

3. **It would be nice to see any impression regarding point sources and non-point sources of emission when applying this model.**

We agree with the Referee that distances over 1 km without ammonia sources are very difficult to achieve in densely agricultural areas and that crop fields in particular pose a challenge. This is especially true during the fertilization season, where these fields can act as very strong ammonia emission sources over a large area. The main difference between these field emissions and barn emissions are that the strong emissions from field fertilization are relatively short lived, decreasing significantly after a few days, while barn emissions are continuous, although the emission strength does vary. As barn emissions can be realistically represented by a small area source (40 x 20 m) with approximately constant emissions, we believed that this type of emission source would be best suited to find a first-order estimate of the blending distance.

Note that the presented simulation framework is well suited to study area emissions from fertilization events, following one of the messages of the manuscript on the potential uses of the simulation framework. One interesting to note is that the emission strength of the field should decrease downwind. As ammonia emission increases the atmospheric concentration downwind, the difference between the atmospheric ammonia and the canopy concentration point is reduced, decreasing the emission strength of the field. The simulation framework is well suited to study such field emissions, after adding a dynamic surface-atmosphere exchange to the simulation framework with a parameterized or fixed compensation point, following the recommendation in Sect. 4.4. However, such an experiment is outside the scope of this study.

Changes made to the revised manuscript
In response to this comment, we added that the simulation framework can be expanded to study the $NH_3$ dispersion from multiple sources or area sources in Sect. 4.5 "Towards an $NH_3$ virtual testbed: integrating fine-scale simulations with advanced observations", line 468.

4. **It would be useful a figure showing the different spatial configuration (emission sources and measuring sites) tested in the simulation with the model**

Following the advice of the Referee, we made a new Figure showing the spatial structure of the percentage change (PC) and blending-distance (BD), which is added to the revised manuscript as Fig. 5 in section 3.2 of the manuscript. In this figure, the white to black color scale shows the percentage change for the fluctuation intensity ($PC_{fI}$) and for the $NH_3$ flux ($PC_F$), similar to the orange dots in Fig. 4a and 4b. The colored contour lines show where the percentage change reaches the three thresholds (5%, 25% & 50%), i.e. the blending-distance. Note each individual grid cell of the model can be considered to be a simulated measurement site, therefore showing different spatial configurations of measurement sites.

The right panel shows this blending-distance for different angles relative to the plume centreline, oriented towards the west (W). Note that in this study, we aim to find the blending-distance in a worst case scenario: the measurement site being located perfectly downwind of the source, at the plume centreline. While the right panel provides the same information as the left panel, it provides new insights in the role of the wind direction and visualizes the uncertainty in the blending-distance calculations, especially for the 5% and 10% threshold levels.

Changes made to the revised manuscript
A new figure, Fig. 5, was added to Sect. 3.2 "Quantitative analysis of the NH3 emission plume impact". At the end of Sect. 3.2, a new paragraph discussing the new figure, starting at line 296.

5. **It would be useful a figure showing information about the weather conditions (temperature, wind speed, wind direction, etc.)**

Changes made to the revised manuscript
Following the advice of the Referee, we added in the first paragraph of Sect. 2.2 (line 126) an additional reference to Figures 2 and 3 by Barbaro at al. (2014). This paper described the meteorology of 8 May 2008 in detail, with Figures 2 and 3 showing vertical profiles and time series of, among other variables, potential temperature, specific humidity, surface fluxes and boundary layer height.

References
Barbaro, E., Vila-Guerau de Arellano, J., Ouwersloot, H. G., Schröter, J. S., Donovan, D. P., and Krol, M. C.: Aerosols in the convective boundary layer: Shortwave radiation effects on the coupled land-atmosphere system, Journal of Geophysical Research: Atmospheres, 119, 505 5845-5863, https://doi.org/10.1002/2013JD021237, 2014.

6. **Effect of background NH3 concentration in the blending-distance: Further discussion on this may be interesting.**

We agree with the Referee and added more discussion on the sensitivity of both $BD_{fI}$ and $BD_F$ to the background molar fraction. For $BD_{fI}$, two processes decrease the magnitude of $PC_{fI}$, resulting in a decreasing $BD_{fI}$ for increasing (initial) background molar fraction. Firstly, the increasing average $NH_{3,bg}$ reduces the fluctuation intensity following Eq. 2 and lowers the relative weight of the $NH_3$ plume fluctuations. Secondly, the increased molar fraction increases the difference in the $NH_3$ air mass characteristics at the top of the boundary layer. This larger difference increases $fI_{bg}$ through entrainment, further reducing the magnitude of $PC_{fI}$ and $BD_{fI}$ with it. For the $NH_3$ flux, the $BD_F$ weak positive correlation is mainly attributed to an increase in the spatial variations in the background $NH_3$ flux, which particularly affect the lower threshold levels (5% and 10%).

Changes made to the revised manuscript
In response to this comment, we added more discussion on the role of $C_{bg}$ to section Section 4.1 ("Uncertainty of the blending-distance estimation"), starting at lines 328 and 377 for $BD_{fI}$ and $BD_F$ respectively.

7. **It would be interesting to establish the minimum requirements that should be accomplished, or to delimit the specific physical context of the site, in which the starting hypothesis and study conclusions are valid.**
   **The replicable capacity of the model and the validity of the minimum distance**

**recommendations concluded may requires much more consideration in the discussion of the results.**

We agree with the Referee that we should have been more clear about the physical context at which the experiments are performed. We therefore added to both the abstract and the conclusions that the experiments are performed over flat homogeneous grassland, centred around the CESAR Observatory at Cabauw. In Sect. 2.2, we added a short description of the surroundings of the CESAR Observatory.

Finally, we added a short discussion on the applicability of the results presented in this study to the end of Sect. 4.2 ("Uncertainty of the blending-distance estimation"). Here, we again mention that the experiments are performed over flat homogeneous grassland and that these results do not necessarily translate to a different physical context. We also stress that the results presented in the manuscript do provide a valuable first-order estimate of the blending-distance. We also want to encourage readers that the concepts and simulation framework are free and with an open source code to be applied to other sites. The code of the adapted DALES model and the scripts used in this study will be made available upon publication.

Changes made to the revised manuscript
In response to this comment, changes were made to lines 11, 122, 414 and 504.

8. **Results and discussion sections are a little bit mixed. The result section contains discussion that may be reallocated in section 4 (i.e lines from 324 to 349).**

We agree with Referee #1 that the results and discussion were a little mixed. This is especially true for former Sect. 3.3, where we not only show the results of the sensitivity study, but also interpret and discuss these results. We therefore moved the former Sect. 3.3 "*Sensitivity of blending-distance to meteorological and NH3 pollution variables*" to the Discussion section, now labelled as Sect 4.1.

9. **Typing errors**
   **L43. Agricultuural --> agricultural**
   **L144 intermittenct --> intermittence**
   **L429. Virutal --> Virtual**

We thank the Referee for pointing out the typing errors. These are corrected in the revised manuscript.

**Response to Referee #2**

1. **It would be nice to add in the manuscript some more discussion about these assumptions of a constant chemical conversion rate which is equally applied to the plume and background NH₃**

We agree with Referee #2 that the implications of our simplified representation of ammonia chemistry should be discussed in the manuscript. Following the advice of the Referee, we added a new paragraph in Sect. 2.1 ("NH₃ turbulent dispersion in DALES"), discussing the role of turbulent mixing on the chemistry of the emission plume. Additionally, a recommendation to improve upon the representation of the NH₃ chemistry is added in Sect. 4.5 ("Towards an NH3 virtual testbed: integrating fine-scale simulations with advanced observations").

Changes made to the revised manuscript
In response to this comment, a discussion on the representation of ammonia chemistry is added to Sect. 2.1, starting at line 82. Furthermore, A few sentences were updated in Sect. (line ) and Sect. 4.5 (line 481).

2. **The scenarios simulated by the authors are analyzed separately. I think it would be nice to have some representation (or at least discussion) of the variability implied by the different scenarios if they are combined, e.g., large emission rate, geostrophic wind and low background simultaneously.**

The main goals of the sensitivity study as presented in the manuscript are to identify the driving variables of the blending-distance and to get an first-order estimate of the range of the blending-distance under these different conditions. Analyzing combined scenarios is not required to reach these goals and would increase the already significant computational costs of the research. However, we do agree with the Referee that it would be interesting to study the blending-distance of combined scenarios to determine the non-linearity of the processes.

To this end, we set up two new experiments where we change 2 variables simulatenously. With the first experiment, we focus on the combined impacts of surface heterogeneity as we vary the strength of the emission source (E) and the surface deposition (D). In the second experiment, we study the combined effect of non-local processes by changing the geostrophic wind speed ($u_g$) and the background NH₃ molar fraction at the start of the analysis window ($C_{bg}$). The values used in each experiment, as well as in the reference experiment (Table 1 in the manuscript), are listed below. With these two experiments, we learn whether there is a linear relation between the results of the sensitivity study presented in the manuscript, i.e. the change in the blending-distance of the combined experiment is equal to the sum of the change in blending-distance of the individual experiments. We therefore define the change in blending-distance as $\Delta BD = BD_{experiment} - BD_{reference}$.

- Combined experiment 1 (CE1):
  - NH₃ emission strength: E = 200 ppb m s$^{-1}$
  - NH₃ deposition strength: D = 0 ppb m s$^{-1}$
- Combined experiment 2 (CE2):
  - Geostrophic wind speed: $u_g$ = 4 m s$^{-1}$
  - Initial NH₃ background molar fraction: $C_{bg}$ = 25 ppb
- Reference experiment:
  - NH₃ emission strength: E = 45 ppb m s$^{-1}$
  - NH₃ deposition strength: D = -0.045 ppb m s$^{-1}$
  - Geostrophic wind speed: $u_g$ = 8 m s$^{-1}$
  - Initial NH₃ background molar fraction: $C_{bg}$ = 10 ppb

In the first combined experiment, we increase the emission strength to 200 ppb m s$^{-1}$ and decrease the surface deposition to 0 ppb m s$^{-1}$ and study the change in the blending-distance for concentration measurements ($BD_{fI}$). Fig. 6 of the manuscript shows that both individual scenarios result in a significant

increase in $BD_{fl}$, so we expect a large increase form the results of CE1. This is indeed shown in Figure **5** below, which shows the blending-distance of each experiment on the left for threshold levels of 50%, 25% and 5%. Here, the reference experiment is shown in black, the two individual scenarios are shown in purple and the combined experiment is shown in orange. The left panel indeed shows that $BD_{fl}$ of CE1 significantly increases for all three threshold levels. The right panel shows the $\Delta BD_{fl}$ for the sum of the individual scenarios in purple and the $\Delta BD_{fl}$ of CE1 in orange. Here, we find that the sum of $\Delta BD_{fl}$ for the individual scenarios is roughly equal to $\Delta BD_{fl}$ of CE1, indicating that there is a linear relation between the blending-distance of these two scenarios.

[Figure]

*Figure VII. $BD_{fl}$ for the reference experiment (black), the individual scenarios (purple) and the combined experiment (CE1) are shown on the left panel for three different threshold levels. The right panel shows the change in blending-distance with respect to the reference experiment ($\Delta BD_{fl}$) for the sum of the individual scenarios (purple) and CE1 (orange).*

In the second combined experiment, we decrease the geostrophic wind speed ($u_g$) to 4 m s$^{-1}$ and increase the background $NH_3$ molar fraction to 25 ppb. Fig. 7 of the manuscript shows that both individual scenarios result in a reduction of $BD_{fl}$, so we expect $BD_{fl}$ of CE1 to be smaller than both the reference and the individual experiments. This is indeed what we see in the left panel of Figure **6**, except for the 50% threshold where CE2 is larger than the $C_{bg}$ = 25 ppb experiment. By analyzing the absolute $\Delta BD_{fl}$ in the right panel, we find that the sum of the individual scenarios is consistently larger than the combination of the two scenarios. This indicates that there is a non-linear relation between the blending-distance of these two scenarios and that, contrary to the results of CE1, the hypothesis is false.

[Figure]

*Figure VIII. $BD_{fl}$ for the reference experiment (black), the individual scenarios (purple) and the combined experiment (CE2) are shown on the left panel for three different threshold levels. The*

*right panel shows the change in blending-distance with respect to the reference experiment (ΔBD$_{fI}$) for the sum of the individual scenarios (purple) and CE2 (orange).*

The results of these two new experiments indicate that combining scenarios is not a straightforward exercise and that more research is required. We believe that such a study is outside the scope of this study and that the analysis of individual scenarios is sufficient to answer the research questions set out in the manuscript.

Changes made to the revised manuscript
No changes were made to the revised manuscript.

3. **Line 227. I think it is misleading to refer to turbulent fluctuations as noise in an observation. I suggest removing the sentence**.

We agree with Referee #2 that our words were chosen poorly in this line and that turbulent fluctuations should not be referred to as noise in an observation. Following the advice of the Referee, we changed the wording of the first and second paragraph of Section 3.1 to improve our message. We removed the mention of noise in an observation and replaced it with "When averaging over 30 minutes, even the large fluctuations between 12:30 and 13:15 are filtered out when averaging over 30 minutes, but these high-frequency turbulent fluctuations could still be present in raw measurement data of high-resolution in-field observations".

Changes made to the revised manuscript
In response to this comment, the second paragraph of Sect. 31 ("Qualitative analysis of the NH$_3$ emission plume impact") is largely rewritten, starting at line 245

4. **Line 235. The authors calculate fI only for values of the mean concertation above a fixed threshold. This is ok but it would be useful to write down how this threshold is significant compared to the local maximum plume mean concentration at the various downwind positions**.

Following the reviewer's comment, we realize when reading Line 235 again that the mention of the threshold of 10$^{-5}$ ppb is out of place in Sect. 3.1. and comes out of the blue. This is an arbitrary threshold, mainly for the purpose of making Fig. 3a in the manuscript. Outside the emission plume, NH$_{3,plume}$ has a value of (very close to) zero, which could lead to an infinitely large fluctuation intensity ($fI = \sigma_{NH_{3,plume}}/\overline{NH_{3,plume}}$). To avoid this from happening, we only calculate the fluctuation intensity for the average NH$_{3,plume}$ higher than an arbitrary threshold, which we set to 10$^{-5}$ ppb. Note that this threshold is not needed when calculating fI for NH$_{3,bg}$ or NH$_{3,total}$ where the molar fraction is never close to zero.

This value is significantly smaller than the average in-plume 3 hour average molar fraction at 37.5 m, peaking at roughly 0.4 ppb as shown in the Figure ***IX***. While the threshold at 10$^{-5}$ ppb is arbitrary, we did test several different thresholds to assure that the emission plume is well captured. Figure ***IX***a and b show the same figure, but with minimum $\overline{NH_{3,plume}}$ thresholds of 10$^{-5}$ ppb (a) and 10$^{-7}$ ppb (b) respectively. The figure shows that reducing this threshold does not significantly widen the plume shown in the Figure, indicating that the emission plume is well captured when using a minimum $\overline{NH_{3,plume}}$ thresholds of 10$^{-5}$ ppb.

Changes made to the revised manuscript
In response to this comment, the mention of the 10$^{-5}$ threshold has been removed from Sect. 3.1 ("Qualitative analysis of the NH$_3$ emission plume impact") and the threshold is now introduced in Sect. 2.3 ("Quantifying the emission plume impact on NH$_3$ measurements") in line 179.

[Figure]

*Figure IX. The average NH$_{3, plume}$, taken between 14:00 and 17:00 CEST. In (a), the limit of the colorbar is set to 10$^{-5}$ ppb, while the colorbar limit is set to 10$^{-7}$ ppb in (b).*

5. **Line 379. Although I think that I understand how the authors give the estimate 6-15km, it would be useful a more detailed explanation for the less acquainted readers**.

Changes made to the revised manuscript
Following the advice of the Referee, we made several changes to subsection 4.3. The original first paragraph of the subsection is split up into two paragraphs and partly rewritten in order to clarify our approach to find a rough estimate from plume dispersion literature.

6. **Line 420-423 are a repetition of the lines 415-418, please remove it.**

We thank the Referee for pointing out the mistake and we removed the repetition.